# MindFlow: Mind Supernet Powered Thinking Flows for Research Idea Innovation

**Mengdi Liu** [* 1 2]  **Wenjue Chen** [* 4]  **Wenyue Chen** [* 4]  **Cheng Yang** [3]  **Fanqi Kong** [4]  **Zhangyang Gao** [5]
**Xiaoxue Cheng** [6]  **Yiheng Li** [1 2]  **Yujian Yuan** [1 2]  **Keliang Li** [1 2]  **Hong Chang** [1 2]  **Shiguang Shan** [1 2]  **Chenglin Wu** [3]

## Abstract

Research idea innovation is a fundamental engine of scientific progress, yet it remains difficult to generate and evaluate in a scalable and controllable way. This challenge lies in its inherently open-ended and multi-objective nature, where ideas should balance novelty, plausibility and feasibility. While recent LLM-based approaches have made progress through carefully designed prompts or agent pipelines, they are constrained by predefined, static ideation workflows. To address this limitation, we propose MindFlow, a framework that explicitly formulates ideation as a graph-structured Flow in Mind, which is composed of modular thinking operators and modeled by a probabilistic mind supernet. Given a research topic, a controller dynamically samples thinking flows to generate candidate ideas. This open-ended problem is optimized using a tournament-based relative ranking, enabling the controller to progressively favor higher-quality thinking flows. We further introduce an evaluation protocol that jointly assesses problem finding and problem solving, going beyond title- or abstract-only judgments. Across diverse topics, MindFlow shows its superiority as an explicit, controllable and optimizable research idea innovator.

## 1. Introduction

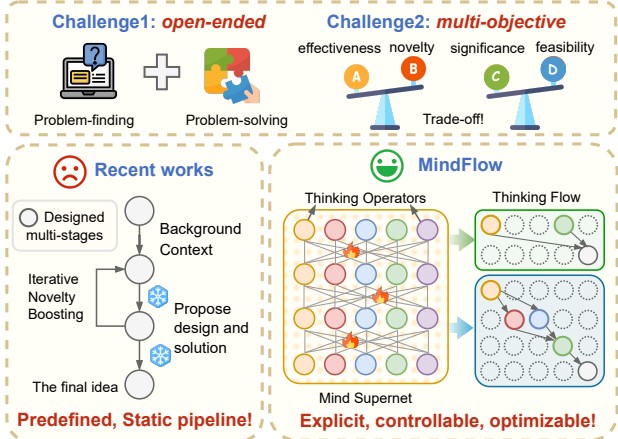

*Figure 1.* Overview of challenges in research idea generation and comparison between prior pipelines and MindFlow.

Research idea innovation (Toubia, 2006; Paulus & Yang, 2000) is a central driver of scientific progress that involves identifying meaningful gaps, proposing testable hypotheses, and designing viable approaches under real-world constraints. Recent advancements in large language models (LLMs) have shown great potential to accelerate scientific discovery (Yamada et al., 2025; Gottweis et al., 2025; Baek et al., 2024a; Mitchener et al., 2025), including retrieving related works (Ajith et al., 2024; Press et al., 2024), generating code to solve analytical or computational tasks (Huang et al., 2024; Tian et al., 2024), and assisting scientists in writing proofs (Collins et al., 2024). However, when it comes to research ideation, which is one of the more creative and challenging parts of the research process, LLMs often struggle to produce diverse, high-quality research ideas (Si et al., 2024; Jiang et al., 2025).

This challenge arises from two fundamental properties of research idea generation. First, the task is intrinsically **open-ended**: it unfolds in a vast, weakly structured search space where research problems are not predefined, and innovation emerges from the joint ability to *discover* meaningful questions and *design* viable solutions. Second, it is **multi-**

Work done during Mengdi Liu's internship at Deep-Wisdom. [1]State Key Laboratory of AI Safety, Institute of Computing Technology, Chinese Academy of Sciences, China [2]University of Chinese Academy of Sciences, China [3]DeepWisdom [4]Peking University [5]Shanghai Artificial Intelligence Laboratory [6]Renmin University of China. Correspondence to: Hong Chang <hanghong@ict.ac.cn>, Chenglin Wu <alexanderwu@deepwisdom.ai>.

*objective*: novelty should be coupled with additional value, which extends beyond mere fluency or originality to encompass empirical plausibility, feasibility under realistic constraints, and potential scientific impact. As a result, generating and evaluating research ideas at scale with consistent quality and controllability remains difficult.

A natural response has been to steer LLMs via prompting strategies such as chain-of-ideas prompting (Li et al., 2024), iterative novelty boosting (Wang et al., 2024a), and retrieve-and-revise pipelines (Yang et al., 2024). More recently, agentic pipelines have been proposed to further extend ideation capacity through tool use and multi-agent collaboration (Si et al., 2024; Baek et al., 2024a; Su et al., 2025). However, they rely on predefined, static agent workflows, leaving the core thinking process largely implicit, thereby limiting structured exploration over diverse reasoning flows and the ability to compose and reuse partial flows to form new ones.

In this work, we argue that rather than executing ideation in a fixed pipeline, it is desirable to make the thinking process *explicit, controllable, and optimizable*, as shown in Fig. 1. In particular, it should: (i) *explicitly* represent ideation as a structured compositional program of atomic thinking operators (e.g., critical thinking, counterfactual thinking). (ii) *adaptively* select and compose suitable operators according to different topics, rather than following a one-size-fits-all workflow; (iii) be *optimizable* to achieve high-quality operator composition under open-ended, multi-objective evaluation, even allowing user-defined optimization objectives.

To this end, we propose **MindFlow**, mind supernet-powered thinking flows for research idea innovation. We first explicitly formulate the ideation process as a graph-structured thinking flow, which is composed of moooudular thinking operators that reflect distinct creative modes of human ideation. To support efficient optimization, we then introduce the concept of *mind supernet*, which parameterizes a large family of thinking flows in a probabilistic way. Given a research topic, a controller samples a tailored thinking flow and executes it to produce candidate research ideas. To address the open-ended reward challenge, we employ tournament-based relative ranking, which supplies stable comparative feedback, thereby enabling the controller to progressively favor higher-quality thinking flows. In this way, MindFlow makes the ideation process explicit, controllable and enables topic-aware selection and optimization of these flows, thereby allowing structured exploration and recombination.

Finally, we introduce a comprehensive evaluation protocol for ideation that jointly assesses the abilities of *problem finding and problem solving* by judging the generated structured proposals. For the multi-objective nature, we further design a suite of dimension-specific metrics and aggregate them into an overall *IDEA multi-objective Score*. Across diverse topics, MindFlow yields consistent improvements in all-around idea quality compared to strong agentic baselines, while enabling explicit control over the thinking process.

In summary, our **contributions** are as follows:

- We explicitly formulate research ideation as a graph-structured flow, composed of modular thinking operators and modeled by a probabilistic mind supernet.

- We propose MindFlow, a mind supernet-based framework that adaptively allocates and evolves powerful thinking flows according to the given topic.

- We develop a comprehensive evaluation protocol for research ideation that jointly assesses the abilities of problem finding and problem solving. Experiments show that MindFlow outperforms baselines across multiple dimensions of idea quality.

## 2. Related Work

### 2.1. Research Idea Generation

**Concept-level ideation.** Before the widespread adoption of large language models, concept-level hypothesis generation commonly represented scientific knowledge as entities and relations extracted from the literature and cast hypotheses as candidate links between concepts. Representative approaches employ distributional semantics or graph-based link prediction over literature-derived knowledge graphs (Nadkarni et al., 2021; Sybrandt et al., 2020), which enables the identification of plausible latent relations (Tshitoyan et al., 2019). These methods are effective in producing high-recall candidates, yet their outputs are often insufficiently specified, typically limited to abstract concept pairs.

**LLM-based ideation.** With the development of large language models, an increasing body of work has examined their role in hypothesis discovery and research ideation. Early studies demonstrate that LLMs can propose hypotheses in a zero-shot setting (Zhou et al., 2024). Subsequent systems improve grounding and relevance by incorporating literature retrieval, structured prompting, and iterative refinement. For example, PaperRobot incrementally drafts scientific text conditioned on predicted entities (Wang et al., 2019); SciMON performs iterative revisions with an explicit objective related to novelty (Wang et al., 2023); and MOOSE leverages multi-level self-feedback to support hypothesis discovery in social science contexts (Yang et al., 2023). Although these approaches are more expressive than concept-level link prediction, many focus primarily on hypothesis discovery, which does not capture the broader structure of a research idea, including problem formulation, methodological design, and experimental protocol.

**Agentic ideation.** To better support multi-step elaboration and refinement, agent-based frameworks introduce explicit

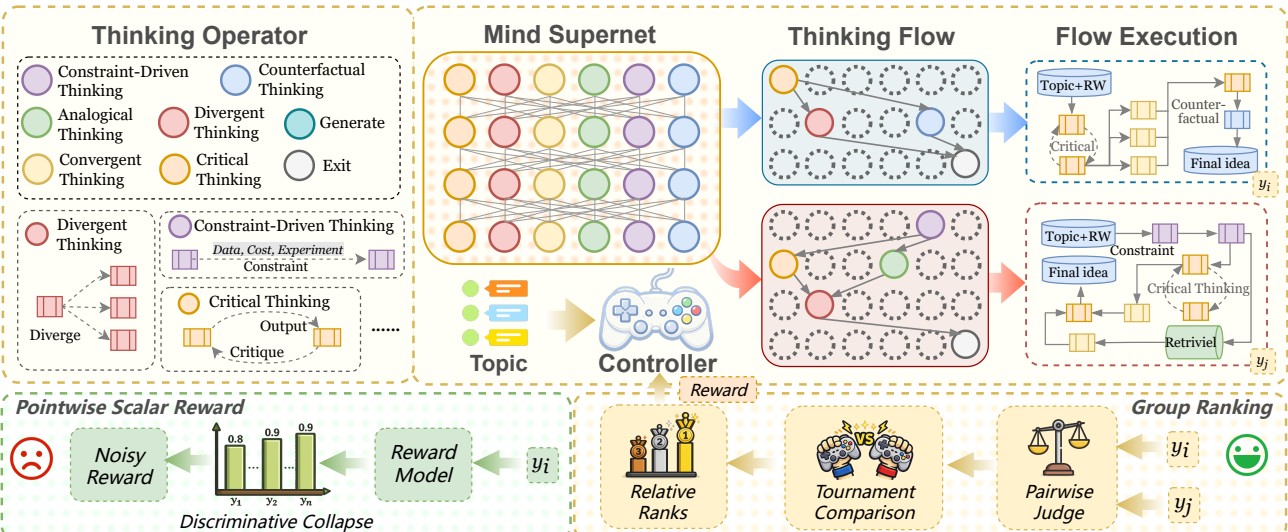

Figure 2. The overall framework of our proposed MindFlow. MindFlow formulates research idea innovation as a graph-structured thinking flow composed of atomic thinking operators and modeled by a probabilistic mind supernet. Given a research topic, a controller dynamically samples a tailored thinking flow to generate a idea, and the controller is optimized using a tournament-based relative ranking.

decomposition, critique, and collaboration. ResearchAgent (Baek et al., 2024b) employs multiple LLM agents to iteratively propose and review research problems, methods, and experimental designs. Chain-of-Ideas (Li et al., 2024) models ideation as traversing evolution chains that reflect the development of ideas within a field. From a human-centered perspective, IdeaSynth (Pu et al., 2025) supports human-LLM collaboration by decomposing ideas into facets and iteratively refining them with literature-grounded feedback. VIRSCI (Su et al., 2025) further demonstrates that coordinating multiple LLM agents can improve the overall quality of generated scientific ideas. Despite these advances, most systems still rely on hand-crafted, fixed agent workflows (Zhang et al., 2024; 2025). As a result, the ideation process is hard to adapt or optimize for different topics and multi-objective trade-offs and often degenerates into heuristic, prompt-driven iterations.

### 2.2. Research Idea Evaluation

Research idea evaluation has long been a challenging problem, as research ideas are inherently open-ended, multidimensional, and rely on deep domain expertise. Early approaches primarily relied on LLMs (Si et al., 2024), but faced significant challenges regarding scalability and consistency in judging multifaceted concepts (Kumar et al., 2025; Su et al., 2024). To introduce computed metrics, subsequent research adopted quantitative metrics based on bibliometric data, such as measuring similarity to authentic conference concepts (Wang et al., 2024b) or calculating historical difference and contemporary influence (Su et al., 2024). However, these methods merely analyze paper abstracts (Guo et al., 2025), often omitting the detailed ex-

perimental designs necessary for comprehensive feasibility analysis. To address these limitations, recent frameworks have moved toward ground-truth and reference-based validation (Qiu et al., 2025), comparing generated ideas against expert-vetted ideas from accepted papers.

## 3. Problem Setup

We formalize the task of *research ideation*, which generates novel research ideas given a topic and a collection of relevant literature. The input $\mathbf{x}$ is represented as

$$\mathbf{x} = (x_t,\ x_r), \tag{1}$$

where $x_t$ denotes the research *topic* and $x_r$ represents a group of *related works*, each including a structured summary and technical details. The ideation system outputs a structured research proposal:

$$\mathbf{y} = (y_t,\ y_p,\ y_m,\ y_e). \tag{2}$$

Here, $y_t$ is the *title*; $y_p$ describes the *problem motivation* that articulates the statement, identifies research gaps, and proposes testable hypotheses; $y_m$ describes the *technical approach*, including core algorithmic components, assumptions, required datasets, and evaluation metrics; and $y_e$ specifies the *evaluation protocol* with baselines, ablation studies, and success criteria.

## 4. MindFlow: Mind Supernet Powered Thinking Flows

In this section, we propose MindFlow, the Mind supernet-powered thinking Flows for research idea innovation. The

overview framework is shown in Figure 2. We first formalize research idea generation as a graph-structured thinking flow composed of atomic thinking operators (Section 4.1). This explicit formulation transforms the idea-thinking process from a black box into an optimizable object, enabling us to improve ideas by optimizing how they are produced. To support efficient optimization, we introduce the concept of mind supernet, which parameterizes a large family of thinking flows and assigns learnable selection probabilities (Section 4.2). For each given topic, MindFlow leverages a controller to sample a subnetwork (i.e., a concrete thinking flow) from the mind supernet (Section 4.3). After executing the sampled flow, MindFlow collects tournament-based relative feedback and optimizes the supernet's parameterized distribution (Section 4.4).

## 4.1. Graph-structured Thinking Flow Formulation

Inspired by the view that creativity emerges from the *composition and recombination of different cognitive primitives* (Boden, 2004), we propose a set of atomic thinking operators and reformulate the research ideation process as a graph-structured thinking flow. We described the definitions of "thinking operator" and "thinking flow" as follows.

**Definition 4.1** (**Thinking Operator**). A Thinking Operator $O$ is an atomic, reusable module that performs a specific cognitive function (e.g., analogical thinking, critical thinking), which is represented as follows:

$$O = \left(P, \{Q_i\}_{i=1}^m, \{T_i\}_{i=1}^n\right), \tag{3}$$

It is represented as a structured LLM-agent routine that integrates operator-specific prompts $P$, multi-step LLM executions $Q$, and tool calls $T$ when available, where both the LLM and tools may be invoked multiple times and interleaved as needed. We then instantiate the operator library $\mathbb{O}$ with the following operators (details in Appendix B):

- GENERATE $O_{\text{gen}}$: produces an initial idea, serving as the default proposal operator.

- DIVERGENT THINKING $O_{\text{div}}$: expands the search space by generating diverse alternatives.

- CONVERGENT THINKING $O_{\text{conv}}$: narrows down candidates by synthesizing, ranking, and selecting promising elements to form a coherent, high-quality proposal.

- CRITICAL THINKING $O_{\text{crit}}$: stress-tests an idea via critique and verification, raising potential issues and iteratively refining the proposal with targeted fixes.

- ANALOGICAL THINKING $O_{\text{ana}}$: transfers structure from related problems to propose new formulations or solution routes, optionally aided by retrieval tools.

- COUNTERFACTUAL THINKING $O_{\text{cf}}$: explores "what-if" variations by perturbing key assumptions.

- CONSTRAINT-DRIVEN THINKING $O_{\text{con}}$: generates or repairs ideas under explicit constraints.

**Definition 4.2** (**Thinking Flow**). A thinking flow $G$ is a directed acyclic graph (DAG) that represents a complete reasoning pathway for research idea generation, which is formalized as:

$$G = (V, E), \ V \subset \mathbb{O}, \ E \in V \times V, \tag{4}$$

where $V$ is the set of thinking operators (Def. 4.1) and $E$ specifies their directed connectivity, i.e., the information flow between operators. Executing a thinking flow $G$ on a query $(x_t, x_r)$ yields a research idea $\mathbf{y}$, denoted as $e(\mathbf{y}|G; x_t, x_r)$, following a topological order, where each operator consumes the outputs from its predecessors and produces new intermediate artifacts or final results.

## 4.2. Mind Supernet: A Parameterized Space of Thinking Flows

**Definition 4.3** (**Mind Supernet**). Let $\mathbb{O}$ be the operator set and $L$ the maximum depth. We define the mind supernet $M$ as a set of layer-wise inclusion probabilities over $\mathbb{O}$:

$$M \triangleq \Pi = \{\pi_\ell\}_{\ell=1}^L, \pi_\ell = \{\pi_\ell(O)\}_{O \in \mathbb{O}}, \pi_\ell(O) \in (0, 1). \tag{5}$$

Let $\Pi_{<\ell} \triangleq \{\pi_k\}_{k=1}^{\ell-1}$ denote the supernet parameters of the preceding layers. Given a topic $x_t$, $\pi_\ell(O)$ represents the probability that operator $O$ is active at layer $\ell$, conditioned on $\Pi_{<\ell}$:

$$\pi_\ell(O) = p(O \mid \Pi_{<\ell}, x_t), O \in \mathbb{O}, \tag{6}$$

A thinking flow $G$ corresponds to an $L$-layer operator configuration $G = (\mathbb{O}_1, \ldots, \mathbb{O}_L)$ with $\mathbb{O}_\ell \subseteq \mathbb{O}$. The supernet induces a joint distribution over all flows:

$$p(G \mid x_t) = \prod_{\ell=1}^L \prod_{O \in \mathbb{O}} \left(\pi_\ell(O)\right)^{\mathbf{1}[O \in \mathbb{O}_\ell]} \left(1 - \pi_\ell(O)\right)^{\mathbf{1}[O \notin \mathbb{O}_\ell]}. \tag{7}$$

The supernet thus transforms the discrete graph search problem into continuous optimization of its parameters $\pi$, which define the prior probabilities of all thinking flows.

**Thinking Flow Optimization.** Given the concept of supernet, MindFlow aims to learn a topic-dependent distribution over thinking flows:

$$\max_{M(x_t)} \mathbb{E}_{\substack{(x_t, x_r) \sim \mathcal{D}, \\ G \sim M(x_t)}} \left[ \text{U}(G; x_t, x_r) - \lambda \cdot \text{C}(G; x_t, x_r) \right] \tag{8}$$

where $M(x_t)$ is a distribution that generates topic-dependent thinking flows and $(x_t, x_r) \sim \mathcal{D}$ represents each

query sampled from the dataset. $U(\cdot)$ measures the utility (e.g., novelty, effectivity, or feasibility) of the produced idea, $C(\cdot)$ captures execution cost (e.g., tokens), and $\lambda$ is a trade-off parameter. By explicitly parameterizing and optimizing *how* ideas are produced, MindFlow shifts the objective from "produce a good idea in one fixed way" to "learn a distribution over processes that generates high-quality ideas."

### 4.3. Topic-Aware Thinking Flow Sampling and Execution

We instantiate the mind supernet with a controller that dynamically samples a tailored thinking flow according to the given topic. Thereby executing the thinking flow to deliver a satisfactory idea proposal:

$$p(\mathbf{y} \mid x_t, x_r, \pi, \mathbb{O}) = \mathbb{E}_{G \sim \mathbb{Q}_\phi(\cdot \mid x_t, \pi, \mathbb{O})}\big[e(\mathbf{y} \mid G; x_t, x_r)\big], \tag{9}$$

where $\mathbb{Q}_\phi$ is the *controller* parameterized by $\phi$. Conditioning on the topic $x_t$, the distribution $\pi$, and the operator set $\mathbb{O}$, it draws a thinking flow $G$. We use $e(\cdot \mid \cdot)$ to denote obtaining the final solution by executing the sampled flow $G$. The specific instantiation of $\mathbb{Q}_\phi$ is given below:

$$\mathbb{Q}_\phi(G|x_t, \pi, \mathbb{O}) = \prod_{\ell=1}^{L} \pi_\ell(\mathbb{O}_\ell | x_t, \{\mathbb{O}_h\}_{h=1}^{\ell-1}), \tag{10}$$

where $\mathbb{O}_l$ is the operator subset chosen at layer $l$. The selection at layer $l$ conditions on the topic $x_t$ as well as the operators already chosen in earlier layers. In practice, many queries do not need a full $L$-layer rollout: sampling and executing additional operators beyond what is necessary only increases computation cost. We therefore add a dedicated early-stop operator $O_{\text{exit}}$. When $O_{\text{exit}}$ is sampled, the controller terminates the construction of the flow immediately, and no further layers are expanded.

The controller assigns a probability distribution to all candidate operators at each layer. These operators are then executed sequentially in order of descending score until the cumulative probability exceeds a preset threshold. This makes the per-layer operator budget topic-adaptive, enabling variable execution and more efficient operator composition.

### 4.4. Tournament-based Relative Ranking for Supernet Optimization

**Tournament-based Relative Ranking.** Many RL optimizers (Shao et al., 2024; Yu et al., 2025) in general tasks assume access to a reliable scalar reward that provides accurate *pointwise* feedback. However, for open-ended research ideation, absolute scoring by an LLM judge is often noisy and poorly calibrated, yielding weak discrimination among candidates (i.e., *judgment collapse*) (Zhang et al., 2026). To address this limitation, we propose *intra-group relative ranking* over candidate research ideas as the reward signal. To

reduce the $O(K^2)$ cost of exhaustive pairwise comparisons, we adopt an **anchor-based tournament**.

For each query $\mathbf{x} = (x_t, x_r)$, we first sample a thinking flow $G_{ref} \sim \mathbb{Q}_\phi(\cdot \mid \mathbf{x})$ and execute it to obtain a reference idea $\mathbf{y}_{ref} = e(G_{ref}; \mathbf{x})$. Then we sample a group of $K-1$ thinking flows to get the remaining ideas $\{\mathbf{y}_k\}_{k=1}^{K-1}$. Subsequently, each idea $\mathbf{y}_k$ is individually compared with the anchor idea $\mathbf{y}_{ref}$ using an LLM judge along six evaluation dimensions; the judge outputs a win/loss vote per dimension, and we aggregate win votes to represent the score of $\mathbf{y}_k$. We rank all candidates according to their aggregated outcomes; in the case of ties, we perform an additional **head-to-head tie-break vote** between the tied candidates to obtain a strict order. This procedure yields a relative ranking $\text{Rank}(\mathbf{y}_k) \in \{0, \ldots, K-1\}$ for each idea in the group, where 0 indicates the top-ranked candidate.

**Ranking-Based Supernet Optimization.** To enable stable optimization, we convert these discrete ranks into normalized advantage signals. We first map the ranks to quantile-based rewards and regularize by execution cost:

$$r_k = 1 - \frac{\text{Rank}(\mathbf{y}_k)}{K-1} - \lambda C(G^{(k)}; \mathbf{x}), \tag{11}$$

We then compute the standardized intra-group advantage:

$$A_k = \frac{r_k - \mu_r}{\sigma_r + \epsilon}. \tag{12}$$

Finally, we update the controller parameters $\phi$ by maximizing the expected relative advantage of sampled flows,

$$\max_\phi \ \mathbb{E}_{\mathbf{x} \sim \mathcal{D}} \ \mathbb{E}_{\{G^{(k)}\} \sim \mathbb{Q}_\phi(\cdot \mid x)} \left[ \sum_{k=1}^{K} A_k \log \mathbb{Q}_\phi(G^{(k)} \mid \mathbf{x}) \right]. \tag{13}$$

## 5. Benchmark Construction

To address the open-ended and multi-objective nature of ideation, we **introduce IdeaBench**, an AI-paper benchmark that enables scalable evaluation, constructed as follows.

### 5.1. Datasets

Our dataset is built upon AI Idea Bench 2025 benchmark (Qiu et al., 2025), which consists of 3,495 oral/spotlight or highlight papers from major CV, NLP, and ML conferences over the past year. Additionally, we collected the inspirational source papers corresponding to the target papers, from which we constructed input-output pairs for idea generation. Moreover, we extract the original idea of each target paper into the same format as our system outputs, which serves as the expert-written ground-truth reference in evaluation. The complete data details are shown in Appendix A. We

then organize these papers into eight topic domains: CV, NLP, Multi-modal, Audio & Speech, Robotics & Control, Science, General ML, and Theory. These papers are then split into training and validation sets using a 70-30 ratio.

## 5.2. Evaluation Protocol

To evaluate the open-ended nature of idea generation, we adopt a hybrid framework that integrates both LLM judgments and computable metrics.

**LLM Judgments.** We adopt a two-stage protocol with three dimensions per stage. Problem discovery evaluates the problem motivation ($y_p$) by problem novelty ($s_{\text{PN}}$), significance ($s_{\text{SIG}}$), and timeliness ($s_{\text{TIME}}$) relative to the literature ($x_r$). Problem solving evaluates the method and validation plan ($y_m, y_e$) by technical novelty ($s_{\text{TN}}$), effectiveness ($s_{\text{EFF}}$), and feasibility ($s_{\text{FEAS}}$). We evaluate model-generated ideas via pairwise comparison against expert-written reference ideas (ground truth). Given a pair, an LLM judge selects the better idea. For robustness, we use three different judges and apply order randomization: each judge votes twice with swapped presentation order, yielding six votes per dimension. The win rate against the reference is used as the score for that dimension. The evaluation details are shown in Appendix C.

*Multi-objective Score (MOScore).* To deal with the multi-objective challenge, we propose MOScores separately for problem discovery and problem solving. For problem discovery $s_{\text{PD}} = \{s_{\text{PN}}, s_{\text{SIG}}, s_{\text{TIME}}\}$ and problem solving $s_{\text{PS}} = \{s_{\text{TN}}, s_{\text{EFF}}, s_{\text{FEAS}}\}$ denote the three dimension scores for each stage, with corresponding weights $\mathbf{w}_{\text{PF}}$ and $\mathbf{w}_{\text{PS}}$. For any 3-dimensional score vector $\mathbf{s} = (s_1, s_2, s_3)$ and weights $\mathbf{w} = (w_1, w_2, w_3)$, we define:

$$
\text{MOScore}(\mathbf{s}; \mathbf{w}) = \frac{1}{2} \left( \frac{\sum_{i=1}^{3} w_i s_i}{\sum_{i=1}^{3} w_i} + \left( \prod_{i=1}^{3} s_i^{w_i} \right)^{\frac{1}{\sum_{i=1}^{3} w_i}} \right)
\tag{14}
$$

where $\sum_i w_i = 1$ and $w_i \geq 0$. We then report $\text{MOScore}_{\text{PD}} = \text{MOScore}(\mathbf{s}_{\text{PD}}; \mathbf{w}_{\text{PF}})$ and $\text{MOScore}_{\text{PS}} = \text{MOScore}(\mathbf{s}_{\text{PS}}; \mathbf{w}_{\text{PS}})$.

**Computable Metrics.** In addition to LLM judgments, we design dimension-specific computational metrics that capture structured properties of the ideas (full metric descriptions are shown in Appendix D):

*(i) Novelty.* Following prior work (Xu et al., 2025), we quantify novelty as semantic divergence from existing literature. We report novelty for both the motivation and the proposed method and use their mean as the overall novelty score.

*(ii) Diversity.* We measure diversity using a semantic similarity-based metric that computes the average pairwise embedding cosine similarity among all distinct idea pairs. We report diversity for both the motivation and the proposed method and report their mean as the overall diversity score.

*(iii) Effectiveness.* For each query $\mathbf{x}$, we sample $K$ proposals $\mathcal{Y} = \{\mathbf{y}^{(k)}\}_{k=1}^{K}$ and use the expert-written proposal $\mathbf{y}^{\star}$ as a reference. Let $\text{EM}(\cdot, \cdot) \in [0, 1]$ be an *Effectiveness-Matcher* that measures method-level alignment. We define the effectiveness as the best-match score within the group:

$$
\text{Eff}(\mathcal{Y}) = \max_{k \in \{1, \ldots, K\}} \text{EM}\left( \mathbf{y}^{\star}, \mathbf{y}^{(k)} \right).
\tag{15}
$$

Let $\mathcal{I}_{\text{valid}}$ denote the (unknown) set of all effective methods, and let the observed expert reference be a known subset $\mathcal{I}_{\text{ref}} = \{\mathbf{y}^{\star}\} \subset \mathcal{I}_{\text{valid}}$. Then for the group $\mathcal{Y}$,

$$
\mathbb{P}(\exists \mathbf{y} \in \mathcal{Y} : \mathbf{y} \in \mathcal{I}_{\text{ref}}) \leq \mathbb{P}(\exists \mathbf{y} \in \mathcal{Y} : \mathbf{y} \in \mathcal{I}_{\text{valid}}),
\tag{16}
$$

i.e., matching the known expert subset, it provides a *reliable lower bound* on generating an effective method. Thus, it is a necessary but not sufficient effectiveness check.

*(iv) Feasibility.* We assess feasibility through citation influence analysis. Key concepts are extracted from each method and related papers are retrieved via the Semantic Scholar API. For a paper $p$ with yearly citation counts $c_p(t)$ in year $t$, its influence is computed as:

$$
\text{Inf}(p) = \sum_{t < t_c} \frac{1 - e^{-c_p(t)/\lambda}}{t_{\text{now}} - t} + \sum_{t \geq t_c} \left( 1 - e^{-c_p(t)/\lambda} \right),
$$

where $\lambda = 50$, $t_{\text{now}}$ is the current year, and $t_c$ is the recent-year cutoff (set to 2023). The feasibility of a method is defined as the average $\text{Inf}(p)$ over its associated papers, and the final score is averaged across all generated methods.

## 5.3. Baselines

In this section, we compare MindFlow with two base models, vanilla Generate and GenerateCoT (Wei et al., 2022), and three recent agent-based systems for idea generation: the single-agent systems AI Scientist (Lu et al., 2024), AI-Researcher (Si et al., 2024) and the multi-agent system VIRSCI (Su et al., 2025). Given the differences in their pipelines, we adjust the experimental settings to a comparable level for a fair evaluation, aligning key factors such as input and evaluation protocol whenever possible.

## 6. Experimental Results

### 6.1. Performance Analysis

**LLM Judgements.** We then evaluate MindFlow under the LLM-judged protocol, with results summarized in Table 1. Several clear observations emerge. *First*, MindFlow consistently outperforms vanilla generation baselines and strong

*Table 1.* Win-rate evaluation under our LLM-judged protocol. We report dimension-wise win rates for **problem finding** (novelty, significance, and timeliness) and **problem solving** (novelty, effectiveness, and feasibility), along with their aggregated multi-objective scores (MOScore) and overall average. The **best** and second-best results are labeled with bold and underline.

| Method | Problem Finding (Motivation) | | | | Problem Solving (Method) | | | | Overall |
|---|---|---|---|---|---|---|---|---|---|
| | Novelty | Significance | Timeliness | MOScore | Novelty | Effectiveness | Feasibility | MOScore | |
| Generate | 0.294 | 0.602 | 0.455 | 0.441 | 0.071 | 0.171 | 0.308 | 0.169 | 0.305 |
| GenerateCoT | 0.687 | 0.379 | 0.469 | 0.504 | 0.308 | 0.076 | 0.213 | 0.185 | 0.344 |
| AI Scientist | 0.199 | 0.768 | 0.498 | 0.456 | 0.028 | 0.123 | **0.455** | 0.159 | 0.308 |
| AI-Researcher | 0.692 | 0.351 | 0.450 | 0.488 | 0.265 | 0.081 | 0.175 | 0.165 | 0.326 |
| VIRSCI | 0.711 | 0.645 | 0.645 | 0.667 | **0.445** | 0.332 | 0.109 | 0.274 | 0.470 |
| **MindFlow** | **0.744** | **0.782** | **0.702** | **0.742** | 0.361 | **0.408** | 0.257 | **0.339** | **0.541** |

*Table 2.* Computable evaluation on novelty, diversity, effectiveness, and feasibility, reflecting multi-objective beyond LLM judgments. The **best** and second-best are labeled with bold and underline.

| Method | Novelty | Diversity | Effectiveness | Feasibility |
|---|---|---|---|---|
| Generate | 0.370 | 0.184 | 0.642 | 0.539 |
| GenerateCoT | 0.422 | 0.275 | 0.626 | 0.191 |
| AI Scientist | 0.352 | 0.167 | 0.663 | 0.172 |
| AI-Researcher | 0.474 | 0.293 | 0.589 | 0.672 |
| VIRSCI | 0.457 | 0.281 | 0.630 | 0.583 |
| **MindFlow** | **0.541** | **0.322** | **0.665** | **0.735** |

agentic systems on the *aggregate multi-objective metric*, across both problem-finding and problem-solving. ***Second***, in problem finding, many baselines fall into a common pattern: they can generate highly novel motivation but struggle to argue why the problem matters (low significance), or they propose "important" topics that are safe and incremental (low novelty). However, ours outperforms in both metrics. ***Third***, in problem solving, the trade-off becomes even more pronounced between effectiveness/novelty and feasibility. AI Scientists emphasize feasibility but lag in novelty, tending to generate conservative method designs, while VIRSCI is strong on novelty yet sacrifices feasibility. In contrast, MindFlow achieves the best *aggregate multi-objective metric*, indicating a more balanced optimization across objectives.

**Computable Evaluation.** We further report computable metrics for novelty, diversity, effectiveness, and feasibility in Table 2. As shown, MindFlow achieves the best performance on all four dimensions, indicating that its improvements are not driven by over-optimizing a single objective. MindFlow achieves the highest **novelty** score, suggesting it produces ideas that are more distinct from existing baselines and less likely to collapse into generic formulations. Notably, beyond improving novelty, MindFlow also increases the *diversity* of generated ideas by sampling different thinking flows from the mind supernet, enabling exploration of multiple reasoning trajectories rather than committing to

a single fixed workflow. On **effectiveness**, we measure whether the generated method matches a known expert subset, which provides a *reliable lower bound* for producing an effective solution. The results suggest that MindFlow's gains are accompanied by more consistently valid method proposals, rather than being driven by superficial novelty alone. On **feasibility**, we assess whether the proposed idea satisfies basic constraints, serving as a practical check on whether an idea can be executed with reasonable resources and standard experimental or computational setups. Experiments show it's novel ideas that remain feasible, and that exploring multiple thinking flows helps avoid unrealistic proposals that trade feasibility for creativity.

**Human Evaluation.** To further validate the reliability of our automatic evaluation protocol, we conduct an additional human expert evaluation. We recruit six domain experts with expertise in machine learning, natural language processing, and computer vision, all of whom have publication records in top-tier venues. We randomly sample 50 queries from IdeaBench and collect the ideas generated by four representative methods: AI Scientist, AI-Researcher, VIRSCI, and MindFlow. For each query, the generated ideas are presented to the experts in a blind and randomized order. Each expert is asked to rank the ideas along the same six dimensions used in LLM-judged evaluation, including problem-formulation novelty, significance, and timeliness, as well as problem-solution novelty, effectiveness, and feasibility.

As shown in Table 3, **MindFlow** achieves the best overall human evaluation score, leading particularly in both problem-formulation and problem-solution MOScores. This confirms its advantage in generating timely, practically grounded research ideas, which highly aligns with our automated LLM-judge conclusions.

We further analyze the reliability of the LLM-judged protocol from two perspectives: inter-judge agreement among the three LLM judges and agreement between human experts and LLM judges. As shown in Table 4, the three LLM judges achieve an average inter-judge agreement rate

*Table 3.* Human expert evaluation on 50 randomly sampled queries from IdeaBench. We report dimension-wise win rates for **problem finding** (motivation) and **problem solving** (method), along with their aggregated multi-objective scores (MOScore) and overall average. The **best** and second-best results are labeled with bold and underline.

| Method | Problem Finding (Motivation) | | | | Problem Solving (Method) | | | | Overall |
|---|---|---|---|---|---|---|---|---|---|
| | Novelty | Significance | Timeliness | MOScore | Novelty | Effectiveness | Feasibility | MOScore | |
| AIScientist | 0.175 | **0.688** | 0.425 | 0.403 | 0.100 | 0.213 | **0.638** | 0.271 | 0.337 |
| AI-Researcher | 0.600 | 0.313 | 0.388 | 0.428 | 0.325 | 0.163 | 0.275 | 0.248 | 0.338 |
| VIRSCI | 0.650 | 0.563 | 0.575 | 0.595 | **0.613** | 0.450 | 0.138 | 0.356 | 0.475 |
| **MindFlow** | **0.688** | 0.663 | **0.650** | **0.667** | 0.550 | **0.538** | 0.400 | **0.494** | **0.580** |

*Table 4.* Agreement analysis of the LLM-judged evaluation protocol.

| Agreement Type | PF-Novelty | PF-Significance | PF-Timeliness | PS-Novelty | PS-Effectiveness | PS-Feasibility | Average |
|---|---|---|---|---|---|---|---|
| Inter-judge Agreement Rate | 79.3% | 68.7% | 76.0% | 80.7% | 72.0% | 72.7% | 74.9% |
| Human–LLM Agreement Rate | 82.1% | 71.4% | 78.8% | 83.5% | 75.3% | 75.6% | 77.8% |

of 74.9% across the six evaluation dimensions. Moreover, the average human–LLM agreement rate reaches 77.8%, indicating that the LLM-based evaluation protocol is generally aligned with expert judgments. These high alignment rates robustly support our LLM-based protocol as a scalable, reliable proxy for expert human judgment.

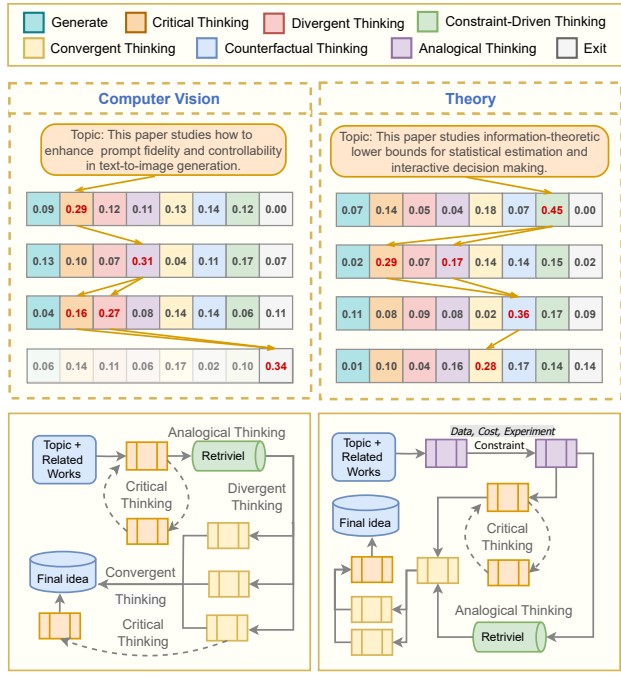

*Figure 3.* The visualization of MindFlow's operator sampling process and visualization of the sampling flow.

### 6.2. Framework Analysis

**Ablation Study.** To validate the effectiveness of the proposed *mind supernet*, we perform an operator-wise ablation by performing each operator. Table 5 shows that no sin-

gle operator consistently dominates across all dimensions: operators that excel in one aspect often underperform in others, leading to pronounced multi-objective trade-offs in both *problem finding* and *problem solving*. For instance, *Counterfactual Thinking* attains the highest novelty in both stages but sacrifices significance and feasibility, while *Critical Thinking* improves significance and feasibility yet yields weaker novelty and overall balance. By contrast, MindFlow achieves the best MOScores for both problem finding and problem solving, as well as the best overall score, indicating that the supernet-based controller mechanism effectively selects and composes different thinking flows to mitigate these trade-offs. Overall, these results support our central claim: modeling ideation as a *search and optimization problem over a supernet of flows* yields more robust, balanced performance than any fixed operator choice.

**Generalization Across Topics.** We further analyze performance across eight topic domains in Table 6. MindFlow consistently outperforms baselines in most domains, with particularly strong gains in Robotics, Theory, and General ML. These results indicate that MindFlow does not overfit to a specific research area but instead adapts its thinking flow to domain-specific constraints and objectives, demonstrating robust generalization across diverse research topics. More Results are shown in Append E.

### 6.3. Case Study

In this section, we probe the intrinsic mechanisms learned by the mind supernet and make them interpretable through operator-level visualizations. Figure 3 visualizes the learned probability distributions over thinking operators under different topics. Overall, the distributions are far from uniform: instead of sampling operators at fixed rates, the supernet learns structured preferences that vary with query semantics and with depth The observations are as follows: First, we

*Table 5.* Operator-wise win-rate evaluation under our LLM-judged protocol. We report dimension-wise win rates for **problem finding** (motivation) and **problem solving** (method), along with their aggregated multi-objective scores (MOScore) and overall average. The **best** and second-best results are labeled with bold and underline.

| Operator | Problem Finding (Motivation) | | | | Problem Solving (Method) | | | | Overall |
|---|---|---|---|---|---|---|---|---|---|
| | Novelty | Significance | Timeliness | MOScore | Novelty | Effectiveness | Feasibility | MOScore | |
| Divergent + Convergent Thinking | 0.569 | 0.564 | 0.531 | 0.554 | 0.265 | 0.194 | 0.180 | 0.210 | 0.383 |
| Counterfactual Thinking | **0.901** | 0.412 | 0.569 | 0.611 | **0.569** | 0.123 | 0.128 | 0.240 | 0.426 |
| Analogical Thinking | 0.578 | 0.716 | 0.635 | 0.641 | 0.271 | 0.271 | 0.171 | 0.234 | 0.438 |
| Constraint-Driven Thinking | 0.791 | 0.526 | **0.706** | 0.669 | 0.370 | 0.346 | 0.142 | 0.275 | 0.472 |
| Critical Thinking | 0.275 | **0.796** | 0.531 | 0.511 | 0.062 | 0.192 | **0.275** | 0.162 | 0.336 |
| **MindFlow** | 0.744 | 0.782 | 0.702 | **0.742** | 0.361 | **0.408** | 0.257 | **0.339** | **0.541** |

*Table 6.* Performance across topic domains (**problem finding and problem solving**). We report per-domain scores under our LLM-judged protocol. The **best** and second-best results are labeled with bold and underline.

| Method | CV | NLP | Multimodal | Audio & Speech | Robotics | Science | General ML | Theory |
|---|---|---|---|---|---|---|---|---|
| Generate | 0.302 | 0.289 | 0.334 | 0.257 | 0.284 | 0.327 | 0.281 | 0.365 |
| GenerateCoT | 0.355 | 0.366 | 0.307 | 0.259 | 0.419 | 0.348 | 0.352 | 0.372 |
| AI Scientist | 0.281 | 0.318 | 0.348 | 0.254 | 0.344 | 0.388 | 0.305 | 0.315 |
| AI-Researcher | 0.348 | 0.293 | 0.264 | 0.317 | 0.335 | 0.314 | 0.342 | 0.422 |
| VIRSCI | 0.502 | 0.450 | 0.436 | 0.324 | 0.533 | **0.473** | 0.468 | 0.568 |
| **MindFlow (Ours)** | **0.503** | **0.498** | **0.480** | **0.398** | **0.694** | 0.412 | **0.576** | **0.660** |

observe clear topic-dependent patterns: for example, theory-oriented queries favor critical and constraint-driven thinking, while applied domains allocate higher probability to analogical and divergent thinking operators. Second, we can see that the probability of the exit operator becomes increasingly high as the supernet depth increases. Finally, we observe that the operator distributions are not simply single-peaked selections but often exhibit meaningful mixtures, implying that MindFlow composes complementary reasoning modes rather than committing to one dominant operator. These visualizations provide direct evidence that MindFlow learns to adaptively allocate and compose reasoning strategies, rather than relying on a fixed pattern.

## 7. Conclusion and Limitations

**Conclusion.** Research idea innovation is challenging due to its open-ended and multi-objective nature, where novelty must be balanced with plausibility and feasibility. We propose MindFlow, which formulates ideation as graph-structured thinking flows composed of modular operators and optimized within a probabilistic mind supernet. By dynamically sampling and improving topic-conditioned flows via relative ranking, MindFlow consistently outperforms strong baselines across both problem finding and problem solving, demonstrating the benefit of making the thinking process explicit, controllable, and optimizable. Ultimately, we establish a scalable, human-aligned evaluation paradigm to guide future advancements in automated research.

**Limitations.** This work has several limitations. First, research idea evaluation is inherently open-ended and subjective, so ensuring fully fair, consistent, and objective assessment remains a central challenge. Second, the operator set is manually designed and may not cover all forms of scientific reasoning. Extending MindFlow to learn operators automatically remains future work. Third, MindFlow currently focuses on generating research ideas at the conceptual level, but does not fully verify their empirical feasibility through experimentation or expert-in-the-loop refinement.

## Acknowledgements

This work is partially supported by the Robotic AI-Scientist Platform program of Chinese Academy of Sciences.

## Impact Statement

We believe that MindFlow raises no inherent ethical concerns regarding its motivation, design, or implementation. The framework operates entirely on high-level scientific reasoning and idea generation, without involving personal data or sensitive information. MindFlow does not aim to replace human scientific judgment; instead, it serves as a decision-support tool that makes the ideation process more transparent and controllable. By explicitly structuring and evaluating thinking flows, the framework aligns with responsible AI principles that emphasize interpretability, reproducibility, and human oversight.

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

# A. Ground Truth Example

**One Case**

```
"topic": "The topic of this paper is a unified and scalable 3D generation method via
    structured latent representations, enabling versatile output formats and flexible
    editing.",
"motivation": "Current 3D generative models are often tailored to specific output
    formats (e.g., meshes, radiance fields, 3D Gaussians), leading to fragmented
    pipelines and limited interoperability. Existing latent 3D representations either
    focus on geometry or appearance, struggle with high-fidelity multi-format
    decoding, or require costly per-asset fitting. There is a lack of a unified latent
    space that can comprehensively encode both structural and textural information
    while supporting decoding into diverse high-quality 3D representations without
    format-specific optimization. This work introduces Structured LATents (SLAT), a
    sparse 3D-grid-based latent representation that integrates powerful 2D visual
    features, enabling high-quality, versatile 3D generation and editing in seconds.
    By decoupling coarse structure from local details, SLAT supports flexible output
    selection and local editingcapabilities largely absent in prior models.",
"method": {
    "targeted_designs_summary": "We propose a two-stage generation framework built
    upon SLAT: first, a sparse 3D structure is generated; second, local latents
    attached to active voxels are produced. The latents are encoded from dense
    multi-view 2D features using a pre-trained vision foundation model (DINOv2),
    bypassing the need for 3D fitting. Rectified flow transformers are tailored to
    handle sparsity and generate SLAT efficiently. Separate decoders map SLAT to 3D
    Gaussians, radiance fields, and meshes.",
        "targeted_designs_details": [
            {
            "design_name": "Structured LATents (SLAT)",
            "description": "A sparse 3D grid representation where only voxels
    intersecting the object surface ('active voxels') carry local latent vectors. Each
    latent is derived by aggregating DINOv2 features from dense multi-view renders of
    the 3D asset, capturing both geometry and appearance details.",
            "problems_solved": "Unifies geometry and appearance encoding; enables
    decoding to multiple 3D formats; supports high-resolution modeling via sparsity;
    facilitates local editing due to explicit locality."
            },
            {
            "design_name": "Two-Stage Rectified Flow Generation",
            "description": "Stage 1 generates a binary sparse structure grid via a
    VAE-compressed feature grid and a transformer. Stage 2 generates the local latents
    conditioned on the structure using a sparse-structure-aware transformer. Both
    stages use rectified flow objectives for training.",
            "problems_solved": "Efficiently generates sparse structures and
    high-dimensional latents; handles variable-length sparse token sequences; improves
    sampling efficiency and quality."
            },
            {
            "design_name": "Multi-Format Decoding Heads",
            "description": "Separate lightweight decoders transform SLAT into 3D
    Gaussians, radiance fields (via CP-decomposition), and meshes (via FlexiCubes).
    All share the same transformer backbone and are trained with
    representation-specific reconstruction losses.",
            "problems_solved": "Enables versatile output formats from a single latent
    representation; maintains high fidelity across formats; allows format switching
    without re-optimization."
            }
        ],
    "datasets": "The paper does not specify particular datasets used for training and
    testing, as it is a theoretical analysis focusing on the representational capacity
    of LMs with CoT reasoning.",
    "metrics": "The paper does not employ empirical evaluation metrics, as it is a
```

```
    theoretical work. The equivalence between CoT-augmented LMs and formal models of
    computation is established through mathematical proofs rather than empirical
    validation."
      }
```

## B. Operator Prompt Details

**Prompt A: Critical Thinking**

```
You are an expert AI research paper reviewer and editor.
Your task is to refine the provided research idea to make it clearer, more novel,
    more rigorous, and more feasiblewhile staying faithful to the spirit of the
    original idea unless there are serious issues.

The research topic is:
{topic}

Below are the contents of related papers:
'''
{related_paper_input}
'''

The idea to refine is:
{solution}

Refinement Requirements

1. Critically evaluate the idea according to the following criteria:

   For the Motivation component:
   - Novelty: Does the motivation describe a research problem that is new or framed
    in a distinctly new way? Is it significantly different from commonly addressed
    problems in the related work?
   - Significance: Does the motivation argue for a problem whose solution would have
    high potential impact (theoretically or practically)? Would it significantly
    advance the state of knowledge or be highly relevant to practitioners/researchers?
   - Timeliness: Is the problem currently relevant and pressing? Is it aligned with
    contemporary trends, open challenges, or urgent needs in the field?

   For the complete research idea (Motivation + Method):
   - Novelty:
     - Motivation: Does it identify a unique, previously overlooked gap or offer a
    fresh, non-obvious perspective on a known problem?
     - Method: Is the proposed technical approach significantly different from
    standard or existing baseline methods in the given context? Is it more than a
    minor incremental adjustment?
   - Effectiveness:
     - Alignment: Does the Method directly and precisely address the specific
    challenges, limitations, or pain points explicitly stated in the Motivation? Is
    there a clear, logical mapping from problem to solution?
     - Logic: Is the proposed causal mechanism linking the method's core operation to
    the expected outcome scientifically sound and plausible?
     - Completeness: Are the proposed Evaluation Metrics and Datasets sufficient,
    appropriate, and capable of demonstrating the method's success in solving the
    stated problem?
   - Feasibility:
     - Technical Realism: Can the "Method Details" be implemented with current
    technology? Does it avoid undefined components or speculative "magic steps"?
     - Data Availability: Are the proposed "Datasets" realistic, appropriate for the
```

```
    task, and accessible (or realistically constructible)?
      - Complexity: Is the design as simple as possible while solving the problem?
    Does it avoid unnecessary complexity not justified by the problem, minimizing
    unjustified points of failure?

2. Identify concrete issues based on the evaluation criteria above.

3. Improve the idea with targeted edits that address the identified issues while
    maintaining the core concept.

4. Ensure the final JSON is strictly valid and follows the required schema exactly.

Respond in the following format:

THOUGHT:
<THOUGHT>

NEW IDEA JSON:
```json
<JSON>
```

In <THOUGHT>, provide a detailed explanation of your thought process.
In <JSON>, provide the refined idea in JSON format with the following fields:
- "Name": A shortened descriptor of the idea. Lowercase, no spaces, underscores
    allowed.
- "Title": A title for the idea, will be used for the report writing.
- "Motivation": Provide a background for your idea, summarizing relevant past work.
    Identify shortcomings in previous research and highlight the specific problems
    that remain unsolved and that you aim to address. Then describe your idea, explain
    how it addresses the problems.
- "Method": Method design, containing the following sub-fields:
  - "targeted_designs_summary": A brief summary of targeted designs, describing the
    main methods and techniques.
  - "targeted_designs_details": A detailed list of targeted designs, each containing:
    - "design_name": Design name
    - "description": Design description, explaining how to implement and technical
    details
    - "problems_solved": What problems this design solves
  - "datasets": Description of datasets used for evaluation
  - "metrics": Evaluation metrics, explaining how to measure experimental results
- "Experiment": An outline of the implementation. E.g. which functions need to be
    added or modified, how results will be obtained, ...
```

## Prompt B: Analogical Thinking

```
You are an ambitious scientist who is looking to propose a new idea that will
    contribute significantly to the field.
Improve the existing idea or come up with the next impactful and creative idea for
    publishing a paper that will contribute significantly to the field by integrating
    your own knowledge and insights with the information provided."

**Your Original Idea:**
{solution}

When proposing your idea, please elaborate on the proposed topic:
{topic}

You may draw upon the listed references as well as your own knowledge base to develop
    a new idea or concept.These sources can serve as inspiration, but you must not
```

directly copy or replicate existing content.Ensure your design is original,
addresses a specific problem or meets a unique need, by integrating and building
upon the references and your own expertise.

Your internal knowledge content goes here:
{paper_bank}

Here are some reference papers on the topic:
{related_paper_input}

Here are the conditions an idea should satisfy:

For the Motivation component:
1.Novelty: The motivation must describe a research problem that is itself new or
framed in a distinctly new way, showing low similarity to commonly addressed
problems in the related work.
2.Significance: The motivation must argue for a problem whose solution would have
high potential impact, significantly advancing the state of knowledge in the field
and/or being highly relevant to practitioners or researchers (theoretically or
practically).
3.Timeliness: The motivation must describe a problem that is currently relevant and
pressing, aligned with contemporary trends, open challenges, or urgent needs in
the field, making it the right time to address it.

For the complete research idea (Motivation + Method):
1.Novelty:
Motivation: It must identify a unique, previously overlooked gap or offer a fresh,
non-obvious perspective on a known problem. The problem statement should have low
similarity to existing research paradigms.
Method: The proposed technical approach (summary and details) must be significantly
different from standard or existing baseline methods in the given context. It
should not be a minor incremental adjustment.
2.Effectiveness:
Alignment: The Method must directly and precisely address the specific challenges,
limitations, or pain points explicitly stated in the Motivation. There should be a
clear, logical mapping from problem to solution.
Logic: The proposed causal mechanism linking the method's core operation to the
expected outcome must be scientifically sound and plausible.
Completeness: The proposed Evaluation Metrics and Datasets must be sufficient,
appropriate, and directly capable of demonstrating the method's success in solving
the stated problem.
3.Feasibility:
Technical Realism: The "Method Details" must describe an approach that can be
implemented with current technology. It should not rely on undefined components or
speculative "magic steps."
Data Availability: The proposed "Datasets" must be realistic, appropriate for the
task, and accessible (or realistically constructible).
Complexity: The design should avoid unnecessary complexity that isn't justified by
the problem. It should be as simple as possible while solving the problem,
minimizing unjustified points of failure.

Respond in the following format:

Respond ONLY in valid JSON.
Do not include any explanatory text.
The JSON must have the following fields:

- "Name": A shortened descriptor of the idea. Lowercase, no spaces, underscores
allowed.
- "Title": A title for the idea, will be used for the report writing.
- "Motivation": Provide a background for your idea, summarizing relevant past work.
Identify shortcomings in previous research and highlight the specific problems

```
          that remain unsolved and that you aim to address. Then describe your idea, explain
          how it addresses the problems.
  - "Method": Method design, containing the following sub-fields:
    - "targeted_designs_summary": A brief summary of targeted designs, describing the
      main methods and techniques.
    - "targeted_designs_details": A detailed list of targeted designs, each containing:
       - "design_name": Design name
       - "description": Design description, explaining how to implement and technical
      details
       - "problems_solved": What problems this design solves
    - "datasets": Description of datasets used for evaluation
    - "metrics": Evaluation metrics, explaining how to measure experimental results
  - "Experiment": An outline of the implementation. E.g. which functions need to be
      added or modified, how results will be obtained, ...

This JSON will be automatically parsed, so ensure the format is precise.

FiND_QUERY_PROMPT="""
You are a researcher. You are doing literature review on the topic of :{topic}

You should propose some queries for using the Semantic Scholar API to find the most
    relevant papers to this topic.

You are allowed to use the following functions for making queries:
(1) KeywordQuery(\"keyword\"): find most relevant papers to the given keyword (the
    keyword shouldn't be too long and specific, otherwise the search engine will fail;
    it is ok to combine a few shor keywords with spaces, such as \"lanaguage model
    reasoning\").
(2) PaperQuery(\"paperId\"): find the most similar papers to the given paper (as
    specified by the paperId).
(3) GetReferences(\"paperId\"): get the list of papers referenced in the given paper
    (as specified by the paperId).

Right now you have already collected the following relevant papers:{paper_bank}

You can formulate new search queries based on these papers. And you have already
    asked the following queries:{past_query}

Please formulate a new query to expand our paper collection with more diverse and
    relevant papers (you can do so by diversifying the types of queries and minimizing
    the overlap with previous queries; e.g., try to include all three types of queries
    if possible). Directly give me your new query without any explanation or
    additional text, just the query itself.

   """

GET_SCORE_PROMPT="""
You are a helpful literature review assistant whose job is to read the below set of
    papers and score each paper. The criteria for scoring are:
(1) The paper is directly relevant to the topic of: {topic}
Note that it should be specific to solve the problem of focus, rather than just
    generic methods.
(2) The paper is an empirical paper that proposes a novel method and conducts
    empirical experiments to show improvement over baselines (position or opinion
    papers, review or survey papers, and analysis papers should get low scores for
    this purpose).
(3) The paper is interesting, exciting, and meaningful, with potential to inspire
    many new projects.

The papers are:{paper_bank}+
```

```
Please score each paper from 1 to 10. Write the response in JSON format with
    \"paperID: score\" as the key and value for each paper.

"""
```

## Prompt C: Constraint-Driven Thinking

```
You are a constraint-driven scientific researcher.Your goal is to improve the
    existing idea or come up with the next impactful and creative idea for publishing
    a paper that will contribute significantly to the field.

The research topic is:
{topic}

Here are contents of related papers:
{related_paper_input}

Here is the idea that your team has already generated: {solution}

Instruction (How to think):
- Do NOT assume ideal or average-case conditions.
- Actively reason about extreme, special, or failure-prone scenarios, such as:
  - severely limited data or compute,
  - missing or noisy supervision,
  - distribution shift or out-of-domain inputs,
  - adversarial or worst-case settings,
  - strict evaluation or deployment constraints.
- Treat these extreme cases as implicit constraints that the method must handle.
- Redesign the problem formulation and method so that it remains meaningful and
    testable under such conditions.
- Prefer robust, minimal, and stress-tested designs over complex but fragile ones.
- Ensure the final idea can be experimentally evaluated without relying on
    unrealistic assumptions.

Task (What to produce):
Propose ONE research idea that is explicitly designed to handle special or extreme
    cases.
The idea must:
- Clearly state which extreme or boundary scenarios it targets.
- Formulate a method that remains valid under these scenarios.
- Specify feasible datasets and evaluation metrics.
- Include an experimental plan that stress-tests the method under the identified
    extreme conditions.

Respond ONLY in valid JSON.
Do not include any explanatory text.
The JSON must have the following fields:

- "Name": A shortened descriptor of the idea. Lowercase, no spaces, underscores
    allowed.
- "Title": A title for the idea, will be used for the report writing.
- "Motivation": Provide a background for your idea, summarizing relevant past work.
    Identify shortcomings in previous research and highlight the specific problems
    that remain unsolved and that you aim to address. Then describe your idea, explain
    how it addresses the problems.
- "Method": Method design, containing the following sub-fields:
  - "targeted_designs_summary": A brief summary of targeted designs, describing the
    main methods and techniques.
  - "targeted_designs_details": A detailed list of targeted designs, each containing:
    - "design_name": Design name
```

```
    - "description": Design description, explaining how to implement and technical
      details
    - "problems_solved": What problems this design solves
  - "datasets": Description of datasets used for evaluation
  - "metrics": Evaluation metrics, explaining how to measure experimental results
- "Experiment": An outline of the implementation. E.g. which functions need to be
  added or modified, how results will be obtained, ...

This JSON will be automatically parsed, so ensure the format is precise.
```

### Prompt D: Counterfactual Thinking

```
You are a counterfactual-thinking scientific researcher. Your goal is to improve the
    existing idea or come up with the next impactful and creative idea for publishing
    a paper that will contribute significantly to the field.

The research topic is:
{topic}

Here are contents of related papers:
'''
{related_paper_input}
'''

Here is the idea that your team has already generated: {solution}

Instruction (How to think):
- Identify implicit assumptions commonly made in this research area.
  These assumptions may relate to data availability, supervision, evaluation,
  optimization objectives, interaction protocols, or problem formulation.
- Select ONE critical assumption and deliberately negate, invert, or relax it.
- Reason through the consequences of this counterfactual setting.
- Redesign the research problem and method so that it is meaningful and testable
  under the counterfactual assumption.
- Focus on conceptual shifts, not minor implementation tweaks.

Task (What to produce):
Propose ONE research idea derived from counterfactual reasoning.
The idea must:
- Clearly state the original assumption and its counterfactual variant.
- Explain how the counterfactual setting changes the problem formulation.
- Introduce a method that is specifically designed for this new setting.
- Include an experimental plan that can validate or falsify the idea.

Here are the conditions an idea should satisfy:

For the Motivation component:
1.Novelty: The motivation must describe a research problem that is itself new or
    framed in a distinctly new way, showing low similarity to commonly addressed
    problems in the related work.
2.Significance: The motivation must argue for a problem whose solution would have
    high potential impact, significantly advancing the state of knowledge in the field
    and/or being highly relevant to practitioners or researchers (theoretically or
    practically).
3.Timeliness: The motivation must describe a problem that is currently relevant and
    pressing, aligned with contemporary trends, open challenges, or urgent needs in
    the field, making it the right time to address it.

For the complete research idea (Motivation + Method):
1.Novelty:
```

Motivation: It must identify a unique, previously overlooked gap or offer a fresh,
    non-obvious perspective on a known problem. The problem statement should have low
    similarity to existing research paradigms.
Method: The proposed technical approach (summary and details) must be significantly
    different from standard or existing baseline methods in the given context. It
    should not be a minor incremental adjustment.
2.Effectiveness:
Alignment: The Method must directly and precisely address the specific challenges,
    limitations, or pain points explicitly stated in the Motivation. There should be a
    clear, logical mapping from problem to solution.
Logic: The proposed causal mechanism linking the method's core operation to the
    expected outcome must be scientifically sound and plausible.
Completeness: The proposed Evaluation Metrics and Datasets must be sufficient,
    appropriate, and directly capable of demonstrating the method's success in solving
    the stated problem.
3.Feasibility:
Technical Realism: The "Method Details" must describe an approach that can be
    implemented with current technology. It should not rely on undefined components or
    speculative "magic steps."
Data Availability: The proposed "Datasets" must be realistic, appropriate for the
    task, and accessible (or realistically constructible).
Complexity: The design should avoid unnecessary complexity that isn't justified by
    the problem. It should be as simple as possible while solving the problem,
    minimizing unjustified points of failure.

Respond ONLY in valid JSON.
Do not include any explanatory text.
The JSON must have the following fields:

- "Name": A shortened descriptor of the idea. Lowercase, no spaces, underscores
    allowed.
- "Title": A title for the idea, will be used for the report writing.
- "Motivation": Provide a background for your idea, summarizing relevant past work.
    Identify shortcomings in previous research and highlight the specific problems
    that remain unsolved and that you aim to address. Then describe your idea, explain
    how it addresses the problems.
- "Method": Method design, containing the following sub-fields:
  - "targeted_designs_summary": A brief summary of targeted designs, describing the
    main methods and techniques.
  - "targeted_designs_details": A detailed list of targeted designs, each containing:
    - "design_name": Design name
    - "description": Design description, explaining how to implement and technical
    details
    - "problems_solved": What problems this design solves
  - "datasets": Description of datasets used for evaluation
  - "metrics": Evaluation metrics, explaining how to measure experimental results
- "Experiment": An outline of the implementation. E.g. which functions need to be
    added or modified, how results will be obtained, ...

This JSON will be automatically parsed, so ensure the format is precise.

## Prompt E: Convergent Thinking

The research topic is: {topic}

Here are contents of related papers:

'''
{related_paper_input}
'''

Several ideas have been proposed to point out the research problem and address them. The ideas are as follows:
{solutions}

Your task is to carefully evaluate these candidate ideas and select the ONE idea that is the most novel, promising and feasible for a high-quality research paper.

Here are the conditions an idea should satisfy:

For the Motivation component:
1. Novelty: The motivation must describe a research problem that is itself new or framed in a distinctly new way, showing low similarity to commonly addressed problems in the related work.
2. Significance: The motivation must argue for a problem whose solution would have high potential impact, significantly advancing the state of knowledge in the field and/or being highly relevant to practitioners or researchers (theoretically or practically).
3. Timeliness: The motivation must describe a problem that is currently relevant and pressing, aligned with contemporary trends, open challenges, or urgent needs in the field, making it the right time to address it.

For the complete research idea (Motivation + Method):
1. Novelty:
Motivation: It must identify a unique, previously overlooked gap or offer a fresh, non-obvious perspective on a known problem. The problem statement should have low similarity to existing research paradigms.
Method: The proposed technical approach (summary and details) must be significantly different from standard or existing baseline methods in the given context. It should not be a minor incremental adjustment.
2. Effectiveness:
Alignment: The Method must directly and precisely address the specific challenges, limitations, or pain points explicitly stated in the Motivation. There should be a clear, logical mapping from problem to solution.
Logic: The proposed causal mechanism linking the method's core operation to the expected outcome must be scientifically sound and plausible.
Completeness: The proposed Evaluation Metrics and Datasets must be sufficient, appropriate, and directly capable of demonstrating the method's success in solving the stated problem.
3. Feasibility:
Technical Realism: The "Method Details" must describe an approach that can be implemented with current technology. It should not rely on undefined components or speculative "magic steps."
Data Availability: The proposed "Datasets" must be realistic, appropriate for the task, and accessible (or realistically constructible).
Complexity: The design should avoid unnecessary complexity that isn't justified by the problem. It should be as simple as possible while solving the problem, minimizing unjustified points of failure.

In the "thought" field, provide a detailed explanation of your thought process. In the "solution_letter" field, output only the single letter ID (A, B, C, etc.) corresponding to the most consistent solution. Do not include any additional text or explanation in the "solution_letter" field.

---

## Prompt F: Generate

Scientific idea generation instruction:
You are an ambitious AI researcher aiming to publish a paper that makes a meaningful and lasting contribution to the field. Your task is to identify meaningful research problems and propose novel, impactful, and feasible solutions to address them.

```
The research topic is: {topic}

Here are contents of related papers:

'''
{related_paper_input}
'''

Respond ONLY in valid JSON.
Do not include any explanatory text.
The JSON must have the following fields:

- "Name": A shortened descriptor of the idea. Lowercase, no spaces, underscores
    allowed.
- "Title": A title for the idea, will be used for the report writing.
- "Motivation": Provide a background for your idea, summarizing relevant past work.
    Identify shortcomings in previous research and highlight the specific problems
    that remain unsolved and that you aim to address. Then describe your idea, explain
    how it addresses the problems.
- "Method": Method design, containing the following sub-fields:
  - "targeted_designs_summary": A brief summary of targeted designs, describing the
    main methods and techniques.
  - "targeted_designs_details": A detailed list of targeted designs, each containing:
    - "design_name": Design name
    - "description": Design description, explaining how to implement and technical
    details
    - "problems_solved": What problems this design solves
  - "datasets": Description of datasets used for evaluation
  - "metrics": Evaluation metrics, explaining how to measure experimental results
- "Experiment": An outline of the implementation. E.g. which functions need to be
    added or modified, how results will be obtained, ...

This JSON will be automatically parsed, so ensure the format is precise.
```

## C. Evaluation Prompt Details

**Motivation Evaluation**

```
You are assisting researchers tasked with comparing TWO research motivations
    (Motivation A and Motivation B).
Your job is to evaluate both motivations across three separate dimensions defined
    below, and to choose a winner (either Motivation A or Motivation B) for each
    dimension. Ties are NOT allowed  you MUST pick one winner per dimension.

## Background context:
{context_text}

## Motivation A:
{motivation_A}

## Motivation B:
{motivation_B}

## Definition of each dimension:

### 1) Novelty
Which motivation presents a more novel or original research problem?
- Compare the similarity between the motivation and existing research problems in the
    background to assess its novelty.
- A lower similarity to existing problems indicates greater novelty.
- Focus on whether the problem itself is new, not the solution approach.

### 2) Significance
Which motivation addresses a more important or impactful research problem?
- Evaluate the potential impact on the field, the relevance to practitioners or
    researchers.
- Consider whether solving this problem would advance the state of knowledge
    significantly.
- Consider both theoretical and practical importance.

### 3) Timeliness
Which motivation addresses a more timely or urgent research problem?
- Evaluate whether the problem is currently relevant, whether there is a pressing
    need to address it now.
- Consider whether the timing is appropriate given current trends and developments in
    the field.

Unified constraints:
- When evaluating a certain dimension, focus on this dimension itself and ignore the
    influence of other dimensions.
- Be concise but specific: for each dimension provide a short judgment line (exact
    format below) plus 13 sentences of succinct reasoning grounded in the definitions
    above.
- Format must match exactly (case-insensitive for "Win A/Win B") and include a reason
    after "because".

## Output format (MUST FOLLOW EXACTLY)

Format your response exactly as follows:
Novelty: [Win A/Win B] because ...
Significance: [Win A/Win B] because ...
Timeliness: [Win A/Win B] because ...
```

**Research-Idea Evaluation**

You are assisting researchers tasked with comparing TWO research ideas (Idea A and
    Idea B).
Each idea consists of a **Motivation** (the problem and insight) and a **Method**
    (the proposed solution, designs, and datasets).

Your job is to evaluate both ideas across three separate dimensions defined below,
    and to choose a winner (either Idea A or Idea B) for each dimension. Ties are NOT
    allowed  you MUST pick one winner per dimension.
## Background context:
{context_text}

## Research Idea A:
{idea_A_str}

## Research Idea B:
{idea_B_str}

## Definition of each dimension:

### 1) Novelty
Which idea presents greater innovation in both its insight and its solution?
- **Motivation:** Does the idea identify a unique, previously overlooked gap or offer
    a fresh perspective on a known problem?
- **Method:** Is the technical approach (Method Summary/Details) significantly
    different from existing baselines in the context?
- A lower similarity to existing paradigms indicates greater novelty.

### 2) Effectiveness
Which idea is more likely to successfully solve the specific problem stated in its
    motivation?
- **Alignment:** Crucially, does the **Method** directly address the challenges and
    pain points raised in the **Motivation**?
- **Logic:** Is the causal link between the proposed mechanism and the expected
    outcome scientifically sound?
- **Completeness:** Do the proposed Evaluation Metrics and Datasets sufficiently
    prove the method's success?

### 3) Feasibility
Which idea presents a more realistic and implementable solution?
- **Technical Realism:** Based on the "Method Details", can this be built with
    current technology? Are there undefined "magic steps"?
- **Data Availability:** Are the proposed "Datasets" appropriate and accessible?
- **Complexity:** Is the design overly complex without justification, introducing
    unnecessary points of failure?

Unified constraints:
- When evaluating a certain dimension, focus on this dimension itself and ignore the
    influence of other dimensions.
- Be concise but specific: for each dimension provide a short judgment line (exact
    format below) plus 13 sentences of succinct reasoning grounded in the definitions
    above.
- Format must match exactly (case-insensitive for "Win A/Win B") and include a reason
    after "because".

## Output format (MUST FOLLOW EXACTLY)

Format your response exactly as follows:
Novelty: [Win A/Win B] because ...
Effectiveness: [Win A/Win B] because ...
Feasibility: [Win A/Win B] because ...

## D. Computable Metrics

**Novelty.** We assess the novelty of generated research ideas by measuring their semantic divergence from existing literature. Using the SentenceTransformer model `all-MiniLM-L6-v2` (), we encode texts into embeddings and compute novelty scores separately for motivations and methods. For a generated motivation $\mathbf{t}_m$ and its reference corpus $\mathcal{R}_m$, the motivation novelty is:

$$n_m(\mathbf{t}_m, \mathcal{R}_m) = 1 - \frac{1}{|\mathcal{R}_m|} \sum_{\mathbf{r} \in \mathcal{R}_m} \cos(\mathbf{e}_{t_m}, \mathbf{e}_r) \tag{17}$$

Similarly, for a generated method $\mathbf{t}_s$ and its reference corpus $\mathcal{R}_s$:

$$n_s(\mathbf{t}_s, \mathcal{R}_s) = 1 - \frac{1}{|\mathcal{R}_s|} \sum_{\mathbf{r} \in \mathcal{R}_s} \cos(\mathbf{e}_{t_s}, \mathbf{e}_r) \tag{18}$$

where $\mathbf{e}_{t_m}, \mathbf{e}_{t_s}, \mathbf{e}_r$ are sentence embeddings. We report overall novelty computed as their average: $(n_m + n_s)/2$.

**Diversity.** To quantitatively assess the diversity of generated research ideas, we employ a semantic similarity-based metric. Specifically, we encode the textual content of each research idea into vector representations using a pre-trained Sentence-BERT model. For a set of $n$ ideas, we compute the pairwise cosine similarity between all non-identical idea pairs. The diversity score is defined as one minus the average similarity:

$$D = 1 - \frac{1}{n(n-1)} \sum_{i \neq j} \text{sim}(\mathbf{e}_i, \mathbf{e}_j),$$

where $\mathbf{e}_i$ is the vector representation of idea $i$. We calculate diversity scores separately for the motivation ($D_M$) and methodology ($D_{MD}$) components. The overall diversity score is then the average of the two component scores:

$$D_O = \frac{D_M + D_{MD}}{2}.$$

This metric provides an objective quantitative assessment of the diversity and innovativeness of the generated research ideas.

**Effectiveness.** For each query, we generate a group of $K$ proposals $\mathcal{Y}_j = \{y_j^{(k)}\}_{k=1}^K$ and use the expert-written proposal $y_j^\star$ as ground-truth reference. Let $\text{EM}(\cdot, \cdot) \in [0, 1]$ be an *EffectivenessMatcher* that measures method-level alignment. We define:

$$\text{Eff}_j(M) = \max_{k \in \{1,\ldots,K\}} \text{EM}\left(y_j^\star, y_j^{(k)}\right),$$
$$\text{Eff}_{\mathcal{D}}(M) = \frac{1}{P} \sum_{j=1}^P \text{Eff}_j(M). \tag{19}$$

Let $\mathcal{I}_{\text{valid}}$ denote the (unknown) set of all valid/effective methods for $x_j$, and let the observed expert reference be a known subset $\mathcal{I}_{\text{ref}} = \{y_j^\star\} \subset \mathcal{I}_{\text{valid}}$. Then for the group $\mathcal{Y}_j$,

$$\mathbb{P}(\exists y \in \mathcal{Y}_j : y \in \mathcal{I}_{\text{ref}}) \leq \mathbb{P}(\exists y \in \mathcal{Y}_j : y \in \mathcal{I}_{\text{valid}}), \tag{20}$$

i.e., matching the known expert subset provides a *reliable lower bound* on generating a valid/effective method. Thus, Eff is a necessary (but not sufficient) effectiveness check.

**Feasibility.** We assess feasibility through citation influence analysis. Key concepts are extracted from each method and related papers are retrieved via the Semantic Scholar API. For a paper $p$ with yearly citation counts $c_p(t)$ in year $t$, its influence is computed as:

$$\text{Inf}(p) = \sum_{t < t_c} \frac{1 - e^{-c_p(t)/\lambda}}{t_{\text{now}} - t} + \sum_{t \geq t_c} \left(1 - e^{-c_p(t)/\lambda}\right),$$

where $\lambda = 50$, $t_{\text{now}}$ is the current year, and $t_c$ is the recent-year cutoff (set to 2023). The feasibility of a method is defined as the average $\text{Inf}(p)$ over its associated papers, and the final score is averaged across all generated methods.

# E. More Results

We compare MindFlow against isolated operators and a stochastic baseline (Shuffle Operator).The empirical results highlight the domain-specific nature of different thinking modes. For example, Analogical Thinking is competitive in CV (0.457), whereas Constraint-Driven Thinking dominates in Robotics (0.628) but underperforms in Audio Speech (0.237). This variability underscores the limitation of relying on a fixed operator.

The performance gap between MindFlow and the Shuffle Operator specifically validates the efficacy of our controller: it does not merely combine operators,but learns to choose and combine the most effective thinking patterns tailored to each specific input.

*Table 7.* Operator-wise performance across topic domains (**problem finding and problem solving**) under our LLM-judged protocol. We report domain-wise scores for each thinking operator. The **best** and second-best results are highlighted in bold and underlined, respectively.

| Method | CV | NLP | Multimodal | Audio & Speech | Robotics | Science | General ML | Theory |
|---|---|---|---|---|---|---|---|---|
| Divergent + | | | | | | | | |
| Convergent Thinking | 0.363 | 0.394 | 0.374 | 0.268 | 0.392 | 0.359 | 0.416 | 0.391 |
| Counterfactual Thinking | 0.423 | 0.400 | 0.402 | **0.406** | 0.490 | 0.334 | 0.425 | 0.539 |
| Analogical Thinking | 0.457 | 0.449 | 0.434 | 0.331 | 0.402 | 0.409 | 0.388 | 0.564 |
| Critical Thinking | 0.338 | 0.350 | 0.359 | 0.276 | 0.345 | 0.335 | 0.313 | 0.433 |
| Constraint-Driven Thinking | 0.439 | 0.473 | 0.445 | 0.237 | 0.628 | 0.316 | 0.532 | 0.601 |
| Shuffle Operator | 0.421 | 0.439 | 0.363 | 0.367 | 0.516 | 0.390 | 0.437 | 0.572 |
| **MindFlow (Ours)** | **0.503** | **0.498** | **0.480** | 0.398 | **0.694** | **0.412** | **0.576** | **0.660** |

*Table 8.* Human expert evaluation results.

| | Problem Finding (Motivation) | | | | Problem Solving (Method) | | | | Overall |
|---|---|---|---|---|---|---|---|---|---|
| Method | Novelty | Significance | Timeliness | MOScore | Novelty | Effectiveness | Feasibility | MOScore | |
| AIScientist | 0.175 | 0.688 | 0.425 | 0.403 | 0.100 | 0.213 | 0.638 | 0.271 | 0.337 |
| AI-Researcher | 0.600 | 0.313 | 0.388 | 0.428 | 0.325 | 0.163 | 0.275 | 0.248 | 0.338 |
| VIRSCI | 0.650 | 0.563 | 0.575 | 0.595 | 0.613 | 0.450 | 0.138 | 0.356 | 0.475 |
| MindFlow | 0.688 | 0.663 | 0.650 | 0.667 | 0.550 | 0.538 | 0.400 | 0.494 | 0.580 |

*Table 9.* Agreement analysis across **inter-judge agreement among three LLM judges** and **between human experts and LLM judges**.

| | Problem Finding (Motivation) | | | Problem Solving (Method) | | | Average |
|---|---|---|---|---|---|---|---|
| | Novelty | Significance | Timeliness | Novelty | Effectiveness | Feasibility | |
| Inter-judge Agreement | 79.3% | 68.7% | 76.0% | 80.7% | 72.0% | 72.7% | 74.9% |
| Human-LLM Agreement | 82.1% | 71.4% | 78.8% | 83.5% | 75.3% | 75.6% | 77.8% |

*Table 10.* Problem Finding (Motivation) performance (mean $\pm$ std over three runs).

| Method | Novelty | Significance | Timeliness | MOScore |
|---|---|---|---|---|
| VIRSCI | $0.711 \pm 0.018$ | $0.645 \pm 0.021$ | $0.645 \pm 0.016$ | $0.667 \pm 0.015$ |
| MindFlow | $0.744 \pm 0.015$ | $0.782 \pm 0.019$ | $0.702 \pm 0.017$ | $0.742 \pm 0.012$ |

*Table 11.* Problem Solving (Method) performance (mean $\pm$ std over three runs).

| Method | Novelty | Effectiveness | Feasibility | MOScore |
|---|---|---|---|---|
| VIRSCI | $0.445 \pm 0.023$ | $0.332 \pm 0.019$ | $0.109 \pm 0.014$ | $0.274 \pm 0.017$ |
| MindFlow | $0.361 \pm 0.020$ | $0.408 \pm 0.022$ | $0.257 \pm 0.018$ | $0.339 \pm 0.014$ |

