# OpenReview forum: "MindFlow: Mind Supernet Powered Thinking Flows for Research Idea Innovation"
_ICML.cc/2026/Conference — ICML 2026 regular_

### Official Review · Reviewer_owFi · 2026-02-24

**Soundness:** 2
**Presentation:** 3
**Significance:** 2
**Originality:** 3
**Overall Recommendation:** 4
**Confidence:** 3

**Summary:**

This paper proposes MindFlow, a framework for research idea generation that makes the “thinking process” explicit and optimizable. Instead of relying on a fixed prompt or a static multi-agent workflow, the authors define an operator library and represent an ideation procedure as a graph-structured thinking flow. They introduce a probabilistic “mind supernet” that parameterizes a large family of such flows via layer-wise operator inclusion probabilities, and a topic-conditioned controller that samples and executes a tailored flow per query.
To optimize the controller/supernet under open-ended, multi-objective evaluation, the paper uses a tournament-based relative ranking scheme: for each topic/literature input, multiple flows generate candidate proposals; an LLM judge compares each candidate against an anchor across several dimensions, producing a relative rank that is converted into an advantage signal for policy-gradient updates. The paper also proposes an evaluation protocol that separates problem finding (novelty/significance/timeliness of the motivation) from problem solving (technical novelty/effectiveness/feasibility of method + evaluation plan), plus additional computable metrics (novelty, diversity, effectiveness proxy via matching a reference, and feasibility proxy via citation influence). Experiments on an “IdeaBench”-style benchmark across multiple AI domains report consistent gains over generation baselines and recent agentic ideation systems, with operator-level ablations and domain breakdowns.

**Compliance With Llm Reviewing Policy:**

Affirmed.

**Final Justification:**

My primary concerns regarding the lack of human evaluation (W1/W5) have been substantially addressed.

**Key Questions For Authors:**

1. Flow structure clarification: How exactly is a “graph-structured thinking flow (DAG)” instantiated during sampling/execution? When multiple operators are chosen in a layer, what is the concrete topology and how are outputs merged/serialized?


2. Controller and training details: What is the controller architecture (e.g., parameterization of ($\pi_\ell(O)$), conditioning on topic and prior selections), and how are per-layer operator “scores” computed for the thresholded execution rule? Please provide full hyperparameters (L, threshold, K, λ), and training stability details.


3. Leakage and benchmark construction: How are “inspirational source papers” selected, and what safeguards ensure the target paper (or near-duplicates) are not included in the related-work input (x_r)? Do you run explicit leakage checks (e.g., title/embedding similarity filters, DOI/paperID exclusion)?


4. Judge robustness: How sensitive are results to the specific LLM judges and prompts? Do you report inter-judge agreement, and do gains hold under different judge families / temperature / prompt variants?


5. Cost/quality tradeoff: You include an execution cost term $(C(\cdot))$. Can you report actual compute/token usage compared to baselines, and show tradeoff curves vs λ (or vs early-exit rate)?

**Limitations:**

The paper acknowledges subjectivity of ideation evaluation and manual operator design, which is good, but it should more directly discuss: (i) risks of LLM-judge optimization and potential “judge overfitting,” (ii) benchmark bias toward reference-matching rather than open-ended novelty, and (iii) potential misuse.

**Strengths And Weaknesses:**

Strengths

- The paper makes a clear and useful move from “prompting/pipelines” to an explicit search/optimization over ideation processes. Modeling ideation as compositional programs of operators is a reasonable abstraction and helps disentangle *how* ideas are produced from *what* ideas are produced. The relative-ranking/tournament approach is a sensible response to noisy absolute scoring for open-ended generation, and the anchor-based design addresses quadratic comparison cost. The framework includes a cost term and an early-exit operator, which is important given that agentic ideation can easily become computationally unbounded.

- The paper is generally well structured: problem setup → operator/flow definition → supernet/controller → optimization → evaluation protocol → experiments/ablations. The separation between problem finding vs. problem solving is a strong design choice and aligns with how human reviewers often judge ideas.

- Scalable, controllable research ideation is an increasingly important problem for LLM-based scientific assistance. The proposed “process-level optimization” view could influence future work on workflow discovery, agentic architecture search, and controllable creativity. Operator-level interpretability (e.g., topic-dependent operator preferences) is appealing as a debugging/control mechanism.

- While individual ingredients resemble ideas from workflow/agent search and preference-based RL, the combination into a probabilistic supernet over cognitive operators and the explicit positioning as “ideation NAS” is novel and clearly articulated. The evaluation protocol (problem finding + solving) is more comprehensive than many “title/abstract-only” ideation evaluations.

Weaknesses / Concerns

- The strongest empirical claims rely heavily on LLM judging. While relative ranking mitigates calibration, the method is still optimizing against a judge that may be gameable or biased toward certain writing styles/structures. The paper would be stronger with either (i) a human expert evaluation subset, or (ii) stronger robustness analyses (judge agreement, sensitivity to judge model/prompt, adversarial checks). The reference-based evaluation (pairwise against expert-written “ground truth” ideas) risks rewarding *similarity to the target reference* rather than truly novel, orthogonal ideas. The paper frames this as a necessary proxy, but it remains unclear how well the benchmark captures real ideation quality rather than “rediscovering” the reference framing.

- The dataset construction (“inspirational source papers” paired with a target paper and an extracted ground-truth idea) is potentially powerful but also raises risks of information leakage or overly strong cues. The paper needs to be extremely explicit about how target papers are excluded from the retrieval set and how “inspirational sources” are selected. Even small leakage would inflate results substantially.

- The formulation says “graph-structured DAG flows,” but the controller sampling is described in layer-wise operator subsets with execution in descending score until a probability threshold is met. It is not fully clear how edges are instantiated, how multiple operators’ outputs are merged, and whether the realized structure is truly a general DAG versus a layered/serialized pipeline. Key implementation details are missing/unclear from the excerpt: controller architecture, how operator “scores” are computed per layer, how intermediate artifacts are represented, how prompts differ by operator in practice (beyond appendices), and the exact training hyperparameters (K in tournament, depth L, threshold, λ cost weight). Reproducibility would also benefit from reporting compute/token budgets and wall-clock cost relative to baselines, especially since cost is in the objective.

- The paper positions itself against “static workflows,” but the boundary with “automated workflow generation / agent architecture search” is close. The novelty would be clearer with a more explicit comparison/discussion of how the proposed supernet differs from existing workflow search methods (and whether their search spaces overlap).

- It is not yet demonstrated that MindFlow generates ideas that are *useful to human researchers* beyond the benchmark. Even a small user study or qualitative analysis of real-world usability (e.g., downstream adoption, novelty judged by domain experts, or success in guiding experiments) would significantly strengthen the contribution.

---

> ### Author Rebuttal · Authors · 2026-03-30
>
> Thank you for your detailed and constructive feedback. We address your concerns below.
>
> > **W1 & W5 & Q4:** human evaluation, robustness analyses, user study, qualitative analysis.
>
> **Reply:** We acknowledge that human expert evaluation is important. We recruited 6 domain experts and randomly selected 50 samples and presented ideas generated by four methods in a randomized order. Each expert was asked to rank those along six dimensions. Besides, we conducted two levels of agreement analysis: inter-judge agreement among three LLM judges (GPT-5, Gemini-2.5-Pro, Claude-4-Sonnet), and agreement between human experts and LLM judges. Results show that human experts’ conclusions are consistent with LLM-judged evaluation. https://anonymous.4open.science/r/MindFlow/Human_evaluation.png
>
> We clarify that our evaluation **does not reward similarity** to the target reference, but identifies samples that **outperform** the expert-written reference.
>
> As shown in the user case study (https://anonymous.4open.science/r/MindFlow/case.png), experts confirm that ours' technical designs and experimental setups are reasonable. MindFlow integrates multiple thinking modes to produce stronger ideas: counterfactual thinking challenges the assumption of seeking a single optimal sequence, analogical thinking transfers particle-based variational inference to form the core solution, and critical/constraint-driven thinking designs a lightweight proxy network and PLM-anchored regularization to ensure feasibility.
>
> > **W2 & Q3:** Clarify benchmark construction and risks of information leakage.
>
> **Reply:** We clarify that there is no risk of information leakage:
>
> - Inspirational papers are selected solely from the reference list of target paper, **without access** to the target paper's core idea. We first use LLM to identify top-10 relevant **references, then manually** select and summarize five representative papers to ensure quality and prevent leakage.
> - The target paper and any duplicates (DOI/arXiv ID/Semantic Scholar ID) are strictly **excluded from retrieval set** by rule.
> - All generation models are released **before** the target papers to avoid model-level leakage.
>
> > **W3 & Q1 & Q2:** Controller, graph-structured DAG flows and training details.
>
> **Reply:** We thank the reviewer for the detailed questions and clarify each point below.
>
> |Concern|Response|
> |--|--|
> |DAG flow sampling (controller architecture)|The controller is a MoE-style network that takes the topic and prior operators as input, outputs activation scores for candidate operators. Current layer’s operators are selected by accumulating scores in descending order until a threshold is reached.|
> |DAG flow execution (information flow)| Within each layer, multiple operators execute in **parallel, with no intra-layer information sharing**. Their outputs are concatenated or aggregated (if has a convergent operator) and passed to the next layer. Layers are connected sequentially, ensuring DAG acyclicity and reducing the search space from exponential to $O(L ·N)$.  |
> |Training hyperparameters|Tournament group size K=5; depth L=4; threshold $τ$ = 0.3; cost weight $λ$=1e-3|
>
> > **W4:** Comparison with Automated Workflow Generation
>
> **Reply:**  While related to automated workflow generation, MindFlow differs fundamentally in both search space and objective:
>
> - **Search Space.** The search space of MindFlow is composed of *cognitive thinking operators*, modeling diverse modes of human ideation. In contrast, prior methods search over general agent actions(e.g., tool use, code execution) which are designed for task completion rather than creative idea generation.
> - **Optimization Objective.** MindFlow optimizes for *open-ended, multi-objective* ideation quality (novelty, significance, timeliness, effectiveness, feasibility) via tournament-based ranking. Existing methods optimize task accuracy on closed benchmarks with predefined ground truth.
>
> > **Q5 & W3-cost:** Cost-quality tradeoff
>
> **Reply:** We provide a detailed cost analysis per query below:
>
> |Method|LLM API Cost ($)|Time (s)|Overall Score|
> |--|--|--|--|
> |AI-Scientist|0.393|98.8|0.308|
> |AI-Researcher|0.285|211.2|0.326|
> |VIRSCI|0.758|116.3|0.470|
> |MindFlow|0.539|99.2|0.541|
>
> Ours achieve the best performance while maintaining lower cost and runtime than VIRSCI (supernet sampling introduces almost no additional time); we also scaled **AI-Researcher** by increasing the number of generated seed ideas.
>
> |Seed ideas num|LLM API Cost ($)|Time (s)|Overall Score|
> |-|-|-|-|
> |50|0.285|211.2|0.326|
> |100|0.502|322.5|0.361|
> |150|0.721|483.8|0.379|
>
> Simply scaling compute cannot close the performance gap, the efficiency of MindFlow comes from both the early-exit operator and the supernet controller that reduce unnecessary LLM calls. We then perform an ablation study of early-exit operator and cost constraint.
>
> |Method|LLM API Cost ($)|Overall Score|
> |-|-|-|
> |MindFlow|0.539|0.541|
> |w/o $O_{exit}$|0.741|0.552|
> |w/o C(·)|0.632|0.564|

---

> > ### Author Rebuttal · Reviewer_owFi · 2026-04-03
> >
> > My primary concerns have been substantially addressed.

---

> > > ### Author Response · Authors · 2026-04-03
> > >
> > > We sincerely thank the reviewer for the very positive follow-up and for raising the rating. We truly appreciate the reviewer’s support and will incorporate the additional results and clarifications into the final version of the manuscript.

---

### Official Review · Reviewer_5Cvv · 2026-03-09

**Soundness:** 3
**Presentation:** 3
**Significance:** 3
**Originality:** 3
**Overall Recommendation:** 3
**Confidence:** 4

**Summary:**

This paper proposes MindFlow, a framework for research idea generation that models ideation as a graph-structured thinking flow rather than a fixed LLM pipeline. By combining a probabilistic Mind Supernet with tournament-based relative ranking, the method adaptively optimizes topic-specific reasoning flows and achieves stronger overall performance than existing baselines.

**Compliance With Llm Reviewing Policy:**

Affirmed.

**Final Justification:**

The paper presents a novel and promising framework for modeling research ideation as an optimizable thinking flow, and the rebuttal meaningfully strengthens the submission with additional cost analysis and human evaluation. However, I remain unconvinced about the reliability of the evaluation framework: the discrepancy between LLM-based and computable metrics is not fully resolved, the definitions of novelty and feasibility remain debatable, and feasibility is still relatively low even under human assessment. The additional validation with coding agents is interesting but indirect, and the evaluation protocol is not yet fully robust. Overall, while the direction is compelling, the current empirical evidence is insufficient to fully support the claims, leading me to a weak reject.

**Key Questions For Authors:**

1.Perhaps I missed it, but the paper does not clearly mention which models were used in the experiments, including both the generation model and the evaluation models.

2.In the paper, only Analogical Thinking appears to involve retrieval tools. I am curious how this component is designed and what its primary role is. Do the other stages not require external references? If so, could hallucination become a serious issue?

3.Are there concrete examples of generated ideas for different methods that could be compared side-by-side? It would also be useful to rule out the possibility that the generated ideas simply match the preference bias of the LLM evaluator, rather than being genuinely useful research ideas.

**Limitations:**

yes

**Strengths And Weaknesses:**

Strengths:

1.Modeling the research innovation process as a dynamically sampled and optimizable thinking flow is a highly novel and powerful concept. It transforms an inherently vague ideation process into a structured problem that can be explicitly defined, searched, and optimized, which represents an important methodological advancement.

2.To address the lack of reliable absolute scores in idea evaluation, the proposed tournament-based relative ranking mechanism is a clever solution. It avoids the issue that LLM judges often provide unstable or collapsed absolute scores, and instead uses more robust relative comparisons to provide effective learning signals. This idea may also be valuable for other reinforcement learning or generative tasks.

Weaknesses:

1.The overall MindFlow pipeline is quite complex. It involves supernet sampling, the execution of multiple thinking operators (each of which may involve multiple LLM calls), and repeated evaluation and optimization via tournament comparisons. This likely introduces substantial computational costs in terms of both token consumption and runtime. The paper does not provide a detailed analysis of these costs (e.g., time cost or token usage), which makes it difficult to assess the practical feasibility of the approach.

2.The evaluation protocol relies entirely on LLM-based judges. It might be helpful to sample a small subset of generated ideas for human evaluation, and analyze the consistency between human judgments and LLM-based evaluation. In addition, using effectiveness, feasibility, and novelty to evaluate the problem-solving capability may not be entirely appropriate, since the proposed framework does not appear to involve actual experimental implementation.

---

> ### Author Rebuttal · Authors · 2026-03-30
>
> Thank you for your detailed and insightful feedback. Below, we address each of your comments and questions:
>
> > **W1:** A detailed analysis of costs.
>
> **Reply:** We provide a detailed cost analysis per query below:
>
> |Method|LLM API Cost ($)|Time (s)|Overall Score|
> |--|--|--|--|
> |AI-Scientist|0.393|98.8|0.308|
> |AI-Researcher|0.285|211.2|0.326|
> |VIRSCI|0.758|116.3|0.470|
> |MindFlow|0.539|99.2|0.541|
>
> Ours achieve the best performance while maintaining lower cost and runtime than VIRSCI (supernet sampling introduces almost no additional time); we also scaled AI-Researcher by increasing the number of generated seed ideas.
>
> |Seed ideas num|LLM API Cost ($)|Time (s)|Overall Score|
> |--|--|--|--|
> |50|0.285|211.2|0.326|
> |100|0.502|322.5|0.361|
> |150|0.721|483.8|0.379|
>
> Even with higher cost and latency, AI-Researcher still falls short of MindFlow, indicating that performance gains cannot be achieved by scaling compute alone.The efficiency of MindFlow comes from both the early-exit operator and the supernet controller that reduce unnecessary LLM calls. We then perform an ablation study of early-exit operator and cost constraint.
>
> |Method|LLM API Cost ($)|Overall Score|
> |-|-|-|
> |MindFlow|0.539|0.541|
> |w/o $O_{exit}$|0.741|0.552|
> |w/o C(·)|0.632|0.564|
>
> > **W2 & Q3:**  Human evaluation study, inter-judge agreement, case study.
>
> **Reply:** We fully acknowledge that human expert evaluation is important. In response, we recruited 6 domain experts (expertize in ML/NLP/CV with publication records at top venues). We randomly selected 50 samples from IdeaBench and presented ideas generated by four methods in a blind, randomized order. Each expert was asked to rank those along the same six dimension.
>
> |Method|PF-Novelty|PF-Significance|PF-Timeliness|PF-MOScore|PS-Novelty|PS-Effectiveness|PS-Feasibility|PS-MOScore|Overall|
> |--|--|--|--|--|--|--|--|--|--|
> |AIScientist|0.175|0.688|0.425|0.403|0.100|0.213|0.638|0.271|0.337|
> |AI-Researcher|0.600|0.313|0.388|0.428|0.325|0.163|0.275|0.248|0.338|
> |VIRSCI|0.650|0.563|0.575|0.595|0.613|0.450|0.138|0.356|0.475|
> |MindFlow|0.688|0.663|0.650|0.667|0.550|0.538|0.400|0.494|0.580|
>
> Notably, the conclusions drawn by human experts are highly consistent with those from LLM-judged evaluation. We then conducted two levels of agreement analysis: inter-judge agreement among the three LLM judges (GPT-5, Gemini-2.5-Pro, and Claude-4-Sonnet), and agreement between human experts and LLM judges.
>
> ||PF-Novelty|PF-Significance|PF-Timeliness|PS-Novelty|PS-Effectiveness|PS-Feasibility|Average|
> |--|--|--|--|--|--|--|--|
> |Inter-judge Agreement Rate|79.3%|68.7%|76.0%|80.7%|72.0%|72.7%|74.9%|
> |Human-LLM Agreement Rate|82.1%|71.4%|78.8%|83.5%|75.3%|75.6%|77.8%|
>
> By side-by-side case study(https://anonymous.4open.science/r/MindFlow/case.png), we can see MindFlow integrates multiple thinking modes to produce a stronger idea: counterfactual thinking challenges the implicit assumption that inverse folding seeks a single optimal sequence; analogical thinking transfers particle-based variational inference from statistical inference to form the core solution; and critical and constraint-driven thinking further introduce designs such as a lightweight proxy network and PLM-anchored regularization to ensure feasibility.
>
> In addition, our ideas include concrete **technical details, benchmark and experimental plans**, enabling assessment of technical soundness and viability. As illustrated in the case study, experts confirm that the technical designs and experimental setups are reasonable. Full implementation of all generated ideas, however, would be prohibitively expensive at scale ($1.2k per paper), which is exactly why idea-level evaluation is necessary before execution.
>
> > **Q1:** Which models were used in the experiments.
>
>  **Reply:** We clarify that the generation backbone is GPT-4o for all methods including baselines, ensuring a fair comparison. We chose GPT-4o which is released prior to the papers in our dataset, as newer models may introduce data leakage from ground-truth papers. For LLM-judged evaluation, we employ three distinct judges GPT-5, Gemini-2.5-Pro, and Claude-4-Sonnet.
>
> > **Q2:** How is Analogical Thinking (with retrieval tools) designed. Do other operators lack external references (lead to hallucination)?
>
>  **Reply:**  Thank you for this insightful question.
>
> - **All operators are grounded on carefully selected literature context**: their inputs already include related works ($x_r$) provided before the thinking flow, so they do not generate ideas without evidence.
> - Introducing retrieval tools in the Analogical Thinking is necessary. The core of analogical thinking is to draw structural ideas from **cross-domain knowledge**, and this design is tailored to its specific needs rather than being required for other operators.
> - **Critical Thinking operator helps mitigate hallucination** by preventing unrealistic or unsupported designs, removing leads to a drop in feasibility (0.257 → 0.175).

---

> > ### Author Rebuttal · Reviewer_5Cvv · 2026-04-03
> >
> > Thank you for your detailed responses—they are very helpful. I have a few follow-up concerns:
> >
> > 1. There is a large discrepancy between LLM-based evaluation and computable metrics (e.g., feasibility, novelty). Does this indicate issues with the evaluation design? Also, defining novelty as dissimilarity to prior work seems questionable—similarity does not necessarily imply lack of novelty, and results may depend heavily on the reference set.
> >
> > 2. The feasibility metric appears to reward ideas that combine popular concepts. However, LLM-generated ideas may superficially merge concepts without real compatibility. Does this risk overestimating feasibility?
> >
> > 3. Feasibility remains relatively low across both human and LLM evaluations. This is concerning, as infeasible ideas have limited value. Have you considered validating some ideas using agentic coding tools (e.g., Claude Code) for partial implementation?
> >
> > 4. Why only compare against ground truth instead of performing pairwise comparisons between methods? Pairwise evaluation might be more robust.

---

> > > ### Author Response · Authors · 2026-04-05
> > >
> > > > **Q1 & Q2:** The discrepancy between LLM-based evaluation and computable metrics. Explanation of computable metrics: feasibility and novelty.
> > >
> > > We thank the reviewer for raising this important concern. We acknowledge that the discrepancy between LLM-based evaluation and computable metrics reflects the inherently open-ended and subjective nature of research ideation: no single metric can fully capture novelty, feasibility, and overall research quality.
> > >
> > > - The computable metrics (novelty and feasibility) are supported by prior literature.
> > >   - Prior works [1,2,3,4,5] all quantify idea novelty based on semantic distance/dissimilarity.
> > >   - Our computable feasibility metric is consistent with [5], while the other works do not provide an explicit computable feasibility metric.
> > > - We acknowledge that the evaluation limitation is real and cannot be fully eliminated. Due to the **open-ended and subjective nature** of this problem, all computable metrics are inherently imperfect approximations and are at best necessary but not sufficient conditions. A more comprehensive, multi-perspective evaluation is necessary precisely because **no single metric is complete**.
> > > - Our multi-dimensional evaluation is a **more comprehensive improvement over prior work:** computable metrics, LLM evaluation, and human evaluation together form a more complete evaluation framework.
> > >
> > > [1] Many heads are better than one: Improved scientific idea generation by a LLM-based multi-agent system. ACL 2025.
> > >
> > > [2] SciMON: Scientific inspiration machines optimized for novelty. ACL 2024
> > >
> > > [3] IdeaBench: Benchmarking Large Language Models for Research Idea Generation. KDD 2025.
> > >
> > > [4] SciPIP: An LLM-based scientific paper idea proposer. arXiv:2410.23166.
> > >
> > > [5] AI Idea Bench 2025: AI research idea generation benchmark. arXiv:2504.14191.
> > >
> > > > **Q3-1:** Feasibility remains relatively low across both human and LLM evaluations. This is concerning, as infeasible ideas have limited value.
> > >
> > > We clarify that feasibility is a **relative score** rather than an absolute one in both human and LLM evaluations. Therefore, a lower feasibility score should not be interpreted as implying “limited value” in an absolute sense. Under our LLM evaluation, feasibility is measured by comparison against the **ground-truth idea**, which represents an already-implemented and peer-reviewed method. Thus a score of 0.5 would indicate that the generated idea is judged as equally feasible as the published ground truth.
> > >
> > > > **Q3-2:** Validate some ideas using agentic coding tools (e.g., Claude Code) for partial implementation.
> > >
> > > Thank you for this constructive comment, and we validate some ideas for partial implementation. We provided the structured proposals (motivation, method, and experiment design) to Claude Code for automated experimental code generation, and found that:
> > >
> > > - Claude Code is able to generate structurally complete and runnable code frameworks, including model definitions, training loops, and evaluation scripts.
> > > - However, the generated code is still difficult to directly reach publication-level quality without human intervention. The main **bottlenecks** include: (1) failure in long-horizon automatic dataset acquisition and preprocessing; (2) incomplete reproduction of baseline methods; (3) the need for large-scale hyperparameter tuning; and (4) reliance on engineering knowledge, such as training stabilization tricks.
> > > - These challenges suggest that current coding agents excel at translating well-specified technical instructions into code, but still struggle with the tacit knowledge and engineering decisions required in research. Addressing this requires developing **specialized research coding agents, which is beyond the scope of this work on idea generation**.
> > >
> > > Nevertheless, the generated code provides additional evidence for feasibility evaluation. We leveraged the code implementations as supplementary context for LLM judges to further assess idea feasibility across different methods, and calculated the agreement score with original LLM evaluations reaches 86.4%.
> > >
> > >
> > > > **Q4:** Compare against ground truth V.S. pairwise comparisons between methods.
> > >
> > > - Comparing against ground truth provides an absolute quality anchor and **ensures fair evaluation**.
> > > - We compare against ground truth for **efficiency, which reduces evaluation cost by about 2.5×**.
> > >
> > > We further supplement the evaluation with a subset-level tournament pairwise evaluation among AI Scientist, AI-Researcher, VIRSCI, and MindFlow.
> > >
> > > |Method|PF-Novelty|PF-Significance|PF-Timeliness|PF-MOScore|PS-Novelty|PS-Effectiveness|PS-Feasibility|PS-MOScore|Overall|
> > > |--|--|--|--|--|--|--|--|--|--|
> > > |AI Scientist|0.050|0.472|0.377|0.253|0.027|0.383|0.661|0.273|0.263|
> > > |AI-Researcher|0.694|0.205|0.238|0.352|0.372|0.183|0.400|0.310|0.331|
> > > |VIRSCI|0.700|0.527|0.605|0.609|0.605|0.611|0.161|0.425|0.517|
> > > |MindFlow|0.555|0.794|0.687|0.675|0.594|0.722|0.377|0.555|0.615|

---

### Official Review · Reviewer_MEbU · 2026-03-10

**Soundness:** 3
**Presentation:** 3
**Significance:** 3
**Originality:** 3
**Overall Recommendation:** 4
**Confidence:** 3

**Summary:**

This paper proposes MindFlow, which formulates research idea generation as graph-structured thinking flows composed of modular operators, parameterized by a probabilistic mind supernet. A controller samples topic-conditioned flows, optimized via tournament-based relative ranking. Experiments on IdeaBench show consistent improvements over baselines across novelty, diversity, effectiveness, and feasibility.

**Compliance With Llm Reviewing Policy:**

Affirmed.

**Final Justification:**

The core idea of this paper is original and well-motivated. The rebuttal also addressed my main concerns with additional experiments, particularly about operator sensitivity and cost. While the manually designed operator remains a limitation, it does not undermine the paper's central contribution. Overall, the originality and experimental results make a reasonable case for the paper and change my score from weak reject to weak accept.

**Key Questions For Authors:**

1. How sensitive is the framework to the specific operator set?
2. Have you considered adding any form of human expert evaluation?
3. What is the total computational cost (LLM API cost, time) of MindFlow compared to the baselines?

**Limitations:**

yes

**Strengths And Weaknesses:**

Strengths:
1. Making ideation explicit, controllable, and optimizable via a supernet search formulation is a creative idea.
2. Tournament-based relative ranking avoids the noise and judgment collapse of absolute LLM scoring, which is a sensible design for open-ended evaluation.
3. Operator-level ablation and topic-level visualization  reveal meaningful patterns, validating the adaptive design.
4. Consistent gains across 8 topic domains show strong generalization.

Weaknesses:
1. The operator set is manually designed and fixed. The paper acknowledges this but does not explore how much the results depend on this particular set.
2. All evaluation relies on LLM judges and computable metrics. There is no human expert evaluation of the generated ideas. Given that the whole point is to generate ideas useful to researchers, this is a notable gap.
3. The computational cost is high (sampling K flows per topic, each with multiple LLM calls, plus tournament comparisons), but no cost analysis or comparison with baselines on cost-effectiveness is provided.

---

> ### Author Rebuttal · Authors · 2026-03-30
>
> Thank you for your detailed and constructive feedback on our work. We appreciate your insights into both the strengths and weaknesses of our approach, and we address each of your points below:
> > **W1 & Q1:** The operator set is manually designed and fixed. How sensitive is the framework to the specific operator set?
>
> **Reply:** We acknowledge that the operator set is manually designed; however, MindFlow is not highly sensitive to the specific operator set. Its performance mainly stems from the combination of diverse cognitive modes, rather than any single operator.
> - **Theory-grounded operators.** The six operators (critical, divergent, convergent, analogical, counterfactual, constraint-driven) are grounded in established cognitive science [1] and cover complementary reasoning modes from exploration to evaluation to integration.
> - **Single-operator ablation reveals multi-objective trade-offs.** Table 3 shows that no single operator performs best across all metrics, with clear novelty–feasibility trade-offs. MindFlow outperforms individual operators, indicating that gains come from composing complementary operators rather than relying on any one of them.
> - **Robust to operator removal.** Removing individual operators does not cause performance collapse, but leads to predictable changes (e.g., removing Critical Thinking reduces feasibility from 0.257 to 0.175). Meanwhile, the supernet adjusts the weights of the remaining operators to compensate, maintaining overall performance (from 0.541 to 0.533).
> - **Extensible and adaptive.** The operator set is not fixed: new operators can be added, and the controller learns to use or ignore them. This makes MindFlow robust to operator variations compared to fixed pipelines.
>
> [1] *Boden, M. A. The creative mind: Myths and mechanisms. Routledge, 2004.*
>
> > **W2 & Q2:** Human evaluation study and inter-judge agreement analysis.
>
> **Reply:** We fully acknowledge that human expert evaluation is important. In response, we recruited 6 domain experts (expertize in ML/NLP/CV with publication records at top venues). We randomly selected 50 samples from IdeaBench, and for each query, we presented ideas generated by four methods in a blind, randomized order. Each expert was asked to rank those along the same six dimensions used in our LLM-judged protocol.
>
> |Method|PF-Novelty|PF-Significance|PF-Timeliness|PF-MOScore|PS-Novelty|PS-Effectiveness|PS-Feasibility|PS-MOScore|Overall|
> |--|--|--|--|--|--|--|--|--|--|
> |AIScientist|0.175|0.688|0.425|0.403|0.100|0.213|0.638|0.271|0.337|
> |AI-Researcher|0.600|0.313|0.388|0.428|0.325|0.163|0.275|0.248|0.338|
> |VIRSCI|0.650|0.563|0.575|0.595|0.613|0.450|0.138|0.356|0.475|
> |MindFlow|0.688|0.663|0.650|0.667|0.550|0.538|0.400|0.494|0.580|
>
> Notably, the conclusions drawn by human experts are highly consistent with those from LLM-judged evaluation. To validate the reliability of our LLM-judged protocol, we conducted two levels of agreement analysis: inter-judge agreement among the three LLM judges (GPT-5, Gemini-2.5-Pro, and Claude-4-Sonnet), and agreement between human experts and LLM judges.
>
> ||PF-Novelty|PF-Significance|PF-Timeliness|PS-Novelty|PS-Effectiveness|PS-Feasibility|Average|
> |--|--|--|--|--|--|--|--|
> |Inter-judge Agreement Rate|79.3%|68.7%|76.0%|80.7%|72.0%|72.7%|74.9%|
> |Human-LLM Agreement Rate|82.1%|71.4%|78.8%|83.5%|75.3%|75.6%|77.8%|
>
>
> > **W3 & Q3:**  Computational cost comparation.
>
> **Reply:** We conducted a cost analysis comparing MindFlow with baselines:
>
> |Method|LLM API Cost ($)|Time (s)|Overall Score|
> |--|--|--|--|
> |AI-Scientist|0.393|98.8|0.308|
> |AI-Researcher|0.285|211.2|0.326|
> |VIRSCI|0.758|116.3|0.470|
> |MindFlow|0.539|99.2|0.541|
>
> Ours achieve the best overall performance while maintaining lower cost and runtime than VIRSCI；we also scaled **AI-Researcher** by increasing the number of generated seed ideas to examine whether simply scaling inference compute can close the performance gap:
>
> |Seed ideas num|LLM API Cost ($)|Time (s)|Overall Score|
> |--|--|--|--|
> |50|0.285|211.2|0.326|
> |100|0.502|322.5|0.361|
> |150|0.721|483.8|0.379|
>
> Even with substantially higher cost and latency, AI-Researcher still falls short of MindFlow, indicating that performance gains cannot be achieved by scaling compute alone. We analysis the efficiency of MindFlow comes from both the early-exit operator and the supernet controller which reduce unnecessary computation and redundant LLM calls.

---

> > ### Author Rebuttal · Reviewer_MEbU · 2026-04-04
> >
> > Thank you for the response. I still have a few questions:
> > 1. The rebuttal cites specific numbers for operator removal, but you didn't provide the corresponding full ablation table and analysis. Table 3 shows single-operator performance, not leave-one-out results. I would also like to see analysis of how computational cost scales with operator set size.
> > 2. In W1/Q1's "extensible and adaptive" part you claimed that new operators can be added and the controller learns to use them, which is not supported by the paper. You already states that "the operator set is manually designed" and that automatic operator learning "remains future work" in the limitations (line 436-437). The controller can select among existing operators, but it does not design new ones.

---

> > > ### Author Response · Authors · 2026-04-06
> > >
> > > > **Q1:**  Full ablation table and analysis of leave-one-out results. Computational cost scales with operator set size.
> > >
> > > **Reply:** Thank you for this valuable suggestion. We provide the corresponding full leave-one-out ablation results below, where each variant removes one operator from MindFlow.
> > >
> > > | Ablation | PF Novelty | PF Significance | PF Timeliness | PF MOScore | PS Novelty | PS Effectiveness | PS Feasibility | PS MOScore | Overall |
> > > |---|---:|---:|---:|---:|---:|---:|---:|---:|---:|
> > > | w/o Counterfactual Thinking | 0.621 | 0.785 | 0.733 | 0.711 | 0.366 | 0.412 | 0.281 | 0.351 | 0.531 |
> > > | w/o Analogical Thinking | 0.643 | 0.679 | 0.706 | 0.676 | 0.432 | 0.247 | 0.278 | 0.314 | 0.495 |
> > > | w/o Constraint-Driven Thinking | 0.672 | 0.779 | 0.633 | 0.693 | 0.361 | 0.239 | 0.241 | 0.278 | 0.485 |
> > > | w/o Critical Thinking | 0.813 | 0.639 | 0.709 | 0.719 | 0.487 | 0.427 | 0.175 | 0.347 | 0.533 |
> > >
> > > We also report how computational cost scales with operator set size:
> > >
> > > | Operator set size | LLM API Cost ($) | Time (s) |
> > > | ----------------- | ---------------- | -------- |
> > > | 2                 | 0.271            | 50.4     |
> > > | 3                 | 0.405            | 74.8     |
> > > | 4                 | 0.489            | 89.6     |
> > > | 5                 | 0.527            | 97.5     |
> > > | 6                 | 0.539            | 99.2     |
> > >
> > > As shown, the cost does not increase linearly. This is because the controller can adaptively select suitable thinking flows through the early-stop operator $O_{exit}$, which terminates flow construction when further expansion is unnecessary.
> > >
> > >
> > >
> > > > **Q2:** In W1/Q1's "extensible and adaptive" part you claimed that new operators can be added and the controller learns to use them, which is not supported by the paper. You already states that "the operator set is manually designed" and that automatic operator learning "remains future work" in the limitations. The controller can select among existing operators, but it does not design new ones.
> > >
> > > We thank the reviewer for pointing out this imprecision and ambiguity in our previous response. We would like to clarify that:
> > >
> > > 1. **The core contribution of MindFlow is learning how to compose operators adaptively, not design new ones.** Our method dynamically selects and combines different thinking operators conditioned on the research topic, and this composition optimization is the main contribution. Automatic operator learning remains future work, as already stated in the limitations.
> > > 2. **The mind supernet framework architecture naturally supports an operator set of arbitrary size.** In the formal definition of the supernet, each layer parameterizes a selection probability over the operator set, which means the framework can support any number of operators without requiring any structural modification.
> > >
> > > We will revise the manuscript to state this point more precisely and avoid conflating **operator composition learning** with **operator discovery**.

---

### Official Review · Reviewer_szc5 · 2026-03-13

**Soundness:** 2
**Presentation:** 3
**Significance:** 3
**Originality:** 4
**Overall Recommendation:** 4
**Confidence:** 3

**Summary:**

MindFlow proposes a new framing for LLM-based scientific ideation. It defines a small library of thinking operators, composes them into a topic-conditioned flow, and models the flow space with a probabilistic mind supernet. A controller samples a flow for a topic, executes it to produce a proposal, and is trained with tournament-style relative ranking. For evaluation, the authors build on the AI Idea Bench 2025 dataset of ~3,500 accepted AI papers across eight domains. The evaluation scores problem finding separately from problem solving. Results show that MindFlow improves the reported overall aggregate over the strongest baseline VIRSCI, mainly by achieving a better trade-off across objectives. Overall, the paper is strong on framing and benchmark design, but the evaluation needs further thought and work.

**Compliance With Llm Reviewing Policy:**

Affirmed.

**Key Questions For Authors:**

1. Am I right in assuming MindFlow spends more compute/tokens on each topic than the baselines? If so, how much do the gains depend on simply spending more tokens?
2. How is the ground truth selected? Can you provide more details about how you obtained the inspirational papers?
3. Can you discuss potential leakage in the construction of IdeaBench?
4. Can you show that by constraining the operators, you can move along the novelty/feasibility trade-off in a predictable manner?
5. How do the metrics you use correlate with human expert judgement?

**Limitations:**

yes

**Strengths And Weaknesses:**

Strengths:
1. The framing of research ideation as a search and optimisation problem is excellent. In particular, the core intuition of modelling and optimising the reasoning process is valuable.
2. The evaluation protocol nicely separates problem finding from problem solving, which is missing in existing approaches.
3. The paper covers multiple experiments, and I particularly like the domain analysis. Very nice!
4. This is a very well-written paper, and I commend the authors for the clarity of the manuscript and the quality of the figures, which is excellent.

Weaknesses:
1. The evaluation is limited to LLM judges and automated metrics. It would be great to see a small-scale human evaluation study where human evaluators judge the generated hypotheses as compared to other methods.
2. I am rather confused about how the information flows inside the supernet. How do the supernet probabilities relate to the controller exactly?
3. The treatment of uncertainty is lacking. Do the results stay the same under multiple runs? It would be beneficial to provide cost, latency, and variance across runs.
4. There is no inter-judge agreement analysis, which would help given the focus on LLM-as-judge evaluation.
5. Some of the automated metrics could be better justified, e.g., the EffectivenessMatcher.
6. The paper makes a point of using tournament-based relative ranking vs scalar-reward optimisation, but it is currently difficult to assess the gains from this choice under the same compute constraint.

---

> ### Author Rebuttal · Authors · 2026-03-29
>
> Thank you for your detailed and constructive feedback. We address your concerns below.
> > **W1 & W4 & Q5:** Human evaluation & inter-judge agreement analysis.
>
> **Reply:** We acknowledge that human expert evaluation is important. We recruited 6 domain experts and randomly selected 50 samples and presented ideas generated by four methods in a randomized order. Each expert was asked to rank those along six dimensions. Besides, we conducted two levels of agreement analysis: inter-judge agreement among three LLM judges (GPT-5, Gemini-2.5-Pro, Claude-4-Sonnet), and agreement between human experts and LLM judges. Results show that human experts’ conclusions are consistent with LLM-judged evaluation. https://anonymous.4open.science/r/MindFlow/Human_evaluation.png
>
> > **W2:**  Explain supernet probabilities/controller/information flows inside the supernet.
>
> **Reply:** The supernet defines a parameterized probability ***distribution*** over thinking flows, while controller performs topic-adaptive, layer-by-layer ***sampling*** from that distribution. During **DAG flow sampling**, controller (MoE-style network) takes the topic and prior operators as input, outputs activation scores for candidate operators. Current layer’s operators are selected by accumulating scores in descending order until a threshold is reached. During **DAG flow execution**, multiple operators are parallel within layer and sequential across layers, ensuring DAG acyclicity and reducing the search space from exponential to $O(L ·|O|)$.
>
> > **W3:** Uncertainty analysis under multiple runs.
>
>  **Reply:** All reported results are the average over three independent runs, we present a subset of results with standard deviation values: https://anonymous.4open.science/r/MindFlow/Uncertainty.png
>
> > **W5:** Justify EffectivenessMatcher
>
>  **Reply:** The EffectivenessMatcher compares the technical components in the generated idea against expert idea. It first uses the Hungarian algorithm to find one-to-one matching between components from the two sides, and then computes the average semantic similarity across all matched component pairs.
>
> > **W6:** Tournament-based relative ranking vs scalar-reward optimisation
>
>  **Reply:** We conducted an ablation comparing tournament-based relative ranking against scalar-reward optimization under the same compute budget.
>
> |Method|PF-Novelty|PF-Significance|PF-Timeliness|PF-MOScore|PS-Novelty|PS-Effectiveness|PS-Feasibility|PS-MOScore|Overall|
> |-|-|-|-|-|-|-|-|-|-|
> |Scalar-reward|0.712|0.582|0.577|0.622|0.346|0.269|0.149|0.247|0.434|
> |Tournament-based Relative Ranking|0.744|0.782|0.702|0.742|0.361|0.408|0.257|0.339|0.541|
>
> > **Q1 & W3-cost**: Cost-quality tradeoff
>
> **Reply:** We provide a detailed cost analysis per query below:
>
> |Method|LLM API Cost ($)|Time (s)|Overall Score|
> |-|-|-|-|
> |AI-Scientist|0.393|98.8|0.308|
> |AI-Researcher|0.285|211.2|0.326|
> |VIRSCI|0.758|116.3|0.470|
> |MindFlow|0.539|99.2|0.541|
>
> Ours achieve the best performance while maintaining lower cost and runtime than VIRSCI (supernet sampling introduces almost no additional time); we also scaled **AI-Researcher** by increasing the number of generated seed ideas.
>
> |Seed ideas num|LLM API Cost ($)|Time (s)|Overall Score|
> |-|-|-|-|
> |50|0.285|211.2|0.326|
> |100|0.502|322.5|0.361|
> |150|0.721|483.8|0.379|
>
> Simply scaling compute cannot close the performance gap, the efficiency of MindFlow comes from both the early-exit operator and the supernet controller that reduce unnecessary LLM calls.
>
> > **Q2**: Ground truth and inspirational papers details.
>
> **Reply:** The **ground truth** is constructed from papers at top conferences over the past year, with LLM-summaried of "title, motivation, technical approach, evaluation" and then verified by human experts. **Inspirational Papers ($x_r$)** are collected from target paper’s reference list, from which an LLM identifies the top-10 most relevant candidates and we manually select and summarize five representative papers.
>
> > **Q3:** Potential leakage of IdeaBench
>
> **Reply:** IdeaBench prevents information leakage via: (1) inspirational papers are selected solely from the reference list of target paper, without access to the target paper's core idea, and are manually verified; (2) the target paper and any duplicates(DOI/arXiv ID/Semantic Scholar ID) are strictly excluded from retrieval set by rule; (3) all generation models are released **before** the target papers, avoiding model-level leakage.
>
> > **Q4:** Move along the novelty/feasibility trade-off in a predictable manner
>
>  **Reply:** Increasing exploration-oriented operators (e.g., Counterfactual Thinking) improves novelty, while constraint-oriented operators (e.g., Critical Thinking) improve feasibility. Varying the Counterfactual prior yields a smooth, predictable trade-off.
>
> |Counterfactual Prior|Novelty(PF)|Feasibility(PS)|
> |-|-|-|
> |0.2|0.275|0.248|
> |0.4|0.370|0.221|
> |0.6|0.412|0.173|
> |0.8|0.456|0.132|
> |1|0.569|0.128|

---

> > ### Author Rebuttal · Reviewer_szc5 · 2026-04-04
> >
> > I thank the authors for their response, which has addressed many of my comments. I have reviewed my score accordingly.

---

> > > ### Author Response · Authors · 2026-04-06
> > >
> > > We thank the reviewer for the feedback and for acknowledging that the concerns have been addressed. We greatly appreciate your time and effort in providing these valuable suggestions, which have helped us strengthen the paper.

---

### Decision · Program_Chairs · 2026-04-30

**Decision:**

Accept (regular)

**Comment:**

MindFlow formulates LLM-based ideation as search over graph-structured thinking flows via a probabilistic supernet over six cognitive operators, trained with tournament-based relative ranking, and introduces a problem-finding vs. problem-solving evaluation protocol. Stengths include the reframing of ideation as an explicit, optimizable process, the tournament ranking as a sensible fix for unstable LLM scoring, and the cleaner evaluation design. The rebuttal added human evaluation with 6 domain experts, inter-judge agreement analysis, cost analysis, leave-one-out ablations, and Claude-Code-based feasibility validation. For camera-ready, authors should fold the rebuttal's ablations, cost analysis, human evaluation, and controller details into the main text, sharpen the operator-composition-vs-discovery framing, and expand limitations to address LLM-judge optimization risks.